# Do opinion leaders know more? Knowledge accuracy, self-confidence, and media use in agricultural issues

Andreas Gabriel [1,2*], Vera Bitsch[3]

**1** Institute for Agricultural Engineering and Animal Husbandry, Bavarian State Research Center for Agriculture, Ruhstorf a.d. Rott, Germany, **2** Horticulture and Food Technology Department, University of Applied Sciences Hochschule Weihenstephan-Triesdorf, Freising, Germany, **3** Chair of Economics of Horticulture and Landscaping, School of Management and School of Life Sciences, Technical University of Munich, Freising, Germany

* andreas.gabriel@hswt.de

## Abstract

This empirical study examines the relationship between knowledge accuracy, self-confidence, and perceived opinion leadership in the context of agricultural issues. Using data from a representative online survey of the German population (n = 2,022), the study analyses how objective factual knowledge and subjective confidence are associated with individuals' self-reported role as opinion leaders, with particular attention to confidence calibration and the Dunning–Kruger Effect (DKE). The theoretical framing integrates Social Cognitive Theory (SCT) and the Two-Step Flow of Communication Model (TSFCM) as heuristic perspectives for understanding social learning and information exchange, while explicitly acknowledging their limitations in contemporary media environments. Across nine factual knowledge statements, the average correct response rate was 64%, while a substantial share of respondents exhibited measurable overconfidence, particularly for false statements. Individuals classified as opinion leaders report significantly higher confidence levels and more intensive use of both, traditional and digital media than the general population, yet they do not demonstrate higher knowledge accuracy. Regression analyses show that confidence and media use are substantially stronger predictors of perceived opinion leadership than factual knowledge. These findings highlight a systematic mismatch between confidence and knowledge among influential communicators in an information-sensitive domain. Rather than demonstrating misinformation effects directly, the results point to the importance of confidence calibration and critical source evaluation in public communication on agriculture and food. The study contributes to the literature by empirically linking confidence miscalibration and opinion leadership within a general population context. From an applied perspective, the findings underline the relevance of targeted communication and literacy initiatives that address overconfidence among highly confident communicators.

**Data availability statement:** All relevant data (codes/labels) are within the manuscript and the Supporting information file.

**Funding:** This research was funded by the Bavarian State Ministry for Food, Agriculture, and Forestry (A/21/17; https://www.stmelf. bayern.de/foerderung/forschungsfoerderung/ index.html). The funding institution had no role in the design of the study; in the collection, analyses of data; in the writing of the article, or in the decision to publish the results.

**Competing interests:** The authors have declared that no competing interests exist.

## 1. Introduction and study objectives

The rapid evolution of information technology and the growing dominance of digital platforms have amplified the role of opinion leaders as pivotal actors in the dissemination of knowledge and shaping public discourse [1,2]. In particular, the agricultural sector presents a unique intersection of complex scientific data, societal concerns (e.g., about sustainability), and the influence of consumer behaviour. In this context, opinion leaders among the general public can act as intermediaries between experts and lay audiences, interpreting and disseminating information in ways that resonate with their followers [3–5]. However, the accuracy and reliability of the knowledge conveyed by opinion leaders have become critical concerns not only because misinformation circulates rapidly in today's media landscape, but also because public expectations toward scientific accuracy have risen over time. Although collective knowledge has improved historically, contemporary communication environments expose individuals to competing, sometimes contradictory information streams that complicate accurate judgement. Previous research demonstrates that social influences significantly affect public understanding in areas such as nutrition, health communication [6,7], climate change [8,9], and genetically modified foods [10–12]. Lack of knowledge and scientific understanding play a role in the denial of climate change [13]. There is also a risk that consumers' media and food literacy will be significantly impaired [6].

Social Cognitive Theory (SCT) offers a robust framework for examining such influences, emphasizing how individuals acquire knowledge and behaviours through the interplay of personal, environmental, and behavioural factors (Fig 1). The figure presents a stylized, post-publication synthesis of confidence calibration patterns commonly associated with the Dunning–Kruger Effect. It is used here as a heuristic illustration rather than as a representation contained in the original Dunning and Kruger (1999) publication. It postulates that learning of individuals occurs through the interaction of (1) personal factors (by cognitive, emotional, and biological events), (2) behavioural patterns, and (3) environmental influences. Key components of SCT include observational learning (individuals learn by observing others), self-efficacy (the belief in one's ability to succeed influences behaviour) and reciprocal determinism (behaviour is shaped by the dynamic interaction between personal and environmental factors) [14]. The theory was developed by Bandura as an extension of classical learning theories and draws attention to the importance of the (social) environment for people's behaviour. In this way, it also suggests intervention methods for changing behavioural determinants [15,16]. To strengthen the theoretical alignment, additional work rooted in social cognitive traditions suggests that group belonging and identity processes shape confidence and information evaluation. Models such as identity fusion [17,18] highlight how strong group alignment can amplify perceived certainty and willingness to communicate publicly [17,18]. This expands the SCT perspective by emphasizing that confidence and communication behaviour may be socially motivated rather than accuracy-driven.

The Dunning-Kruger Effect (DKE), first identified by psychologists David Dunning and Justin Kruger in 1999, refers to a cognitive bias where individuals with low

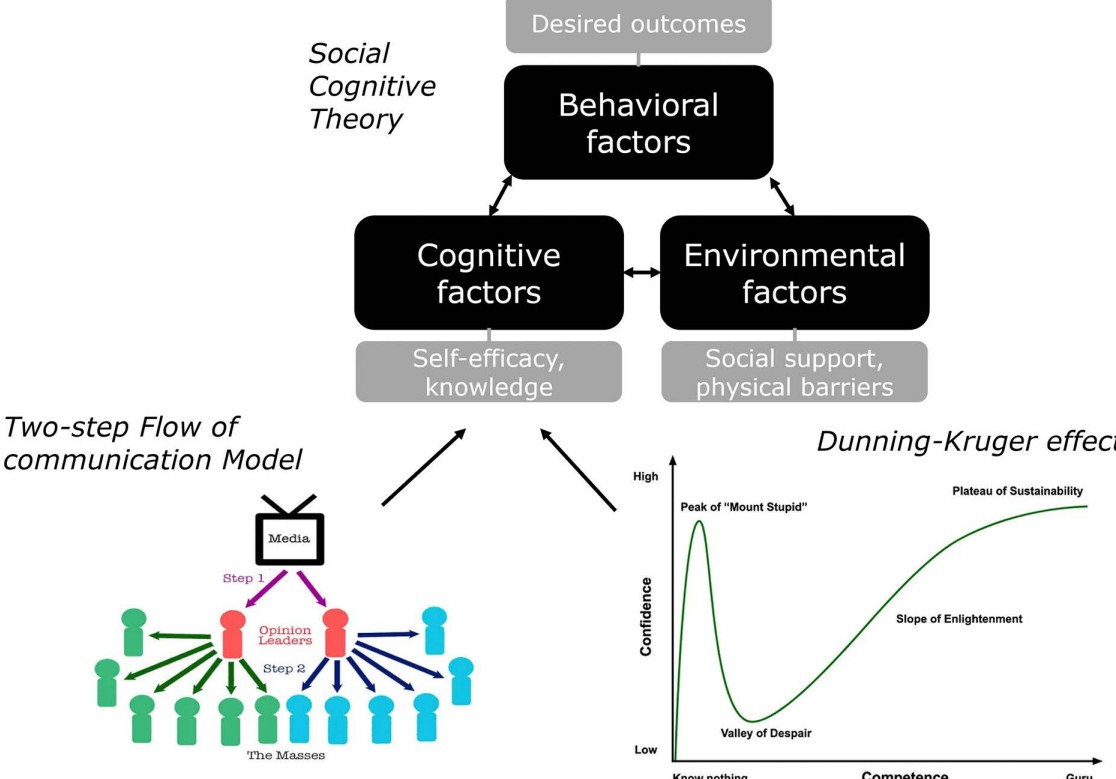

**Fig 1. Integration of two-step flow of communication model (left), and Dunning-Kruger effect (right) into the cognitive component of social cognitive theory (adapted from [14]).**

competence in a particular domain overestimate their abilities, while those with high competence tend to underestimate their own expertise [19]. This phenomenon arises because the skills needed to perform well in a given area are often the same skills required to accurately assess one's performance [20]. Further, when individuals lack awareness of their own incompetence, they fail to recognize the necessity for self-improvement [21]. Consequently, those who lack knowledge or expertise also lack the ability to recognise their own shortcomings [22]. The effect can be illustrated graphically using the relationship between self-confidence and competence. People with limited knowledge often overestimate their abilities, leading to a high initial peak of self-confidence known as the peak of 'Mount stupid' (Fig 1). As competence and knowledge increase, self-confidence usually drops to the 'Valley of despair' where one realises one's limitations. Over time, with further learning and more experience, self-confidence gradually rises along the 'Slope of enlightenment' and stabilises at the 'Plateau of sustainability,' reflecting a realistic understanding of one's expertise.

However, Fig 1 is not part of the original DKE publication. It has become a commonly used post-publication illustration of confidence–competence relationships. To avoid misunderstandings, we contextualise the figure accordingly and refrain from using potentially pejorative labels.

Moreover, the DKE can be integrated with broader communication theories. [23,24] suggests that people evaluate information in ways that protect identity and group belonging, rather than seeking accuracy. This aligns with evidence from Spiral of Silence [25] and related concepts, which highlight that agreement, heuristic cues and social acceptance often outweigh factual correctness when individuals decide whose opinions they follow. Related research on market mavenship demonstrates that self-confidence and communicative activity—rather than topic-specific expertise—often drive perceived

influence [26,27]. This literature is closely aligned with opinion leadership research and helps contextualise our operationalization of the opinion leadership index.

Opinion leaders can take on a special role in their social environment. An opinion leader is an individual who has significant influence over the attitudes, beliefs, and behaviours of others within a specific social group or community [5,28,29]. They are typically seen as knowledgeable, credible, and trustworthy sources of information in their areas of expertise or interest [4]. There is plenty of evidence that consumers trust word-of-mouth recommendations from friends and relatives more than commercial information sources [30,31]; this holds especially for those interpersonal sources with strong ties [32,33]. In the context of communication theories, the Two-Step Flow Communication Model (TSFCM), initially proposed by Katz and Lazarsfeld [34], opinion leaders play a pivotal role in shaping public opinion and behaviour (Fig 1). They amplify the impact of messages by tailoring them to the values, needs, and understanding of their audiences, making the information more relevant and actionable. The TSFCM proposes that information flows from mass media to opinion leaders, who then interpret and disseminate it to others in their social network. This aligns with SCT in the way that opinion leaders may serve as social models, influencing others' understanding and behaviour regarding specific topics, consistent with SCT's focus on observational learning. This means that observers not only learn from media directly but also from the mediated interpretations provided by opinion leaders [31,35]. Then, as SCT acknowledges the power of social structures in shaping learning, the TSFCM emphasizes how information filtered through opinion leaders becomes more personally relevant and impactful for individuals. Lastly, opinion leaders can bolster the self-efficacy of others by framing information in a way that encourages action or belief in personal capabilities [36,37].

For instance, when opinion leaders advocate for healthier nutrition choices, they shape individuals' confidence in their ability to adopt these behaviours [6,38]. Thus, opinion leaders, according to TSFCM, serve as key nodes in the social network, amplifying the reach and impact of mass media content. The original TSFCM was lately further evolved into the Multi-Step Flow Model to reflect the complexities of modern communication networks and social media [39]. In this expanded model, information flows through multiple layers of opinion leaders rather than a single intermediary stage. Social media influencers play a pivotal role, acting as digital opinion leaders who curate, interpret, and disseminate information to their followers [40]. These multipliers frequently enhance and propagate messages from traditional media and other sources, fostering a dynamic, bidirectional flow of communication. At a second level, this information is absorbed and shared through peer and friend interactions.

However, TSFCM alone cannot explain influence in complex, polarized environments where individuals select from competing information nodes. Contemporary opinion leadership often emerges in multi-directional, identity-driven ecosystems. We therefore frame TSFCM as a foundational but not exhaustive model, complemented by DKE, motivated reasoning, and modern communication theories. Both, the TSFCM and the DKE complement SCT by providing insights into how social influences and cognitive biases interact during the learning process of individuals. They enrich the understanding of how individuals acquire knowledge, perceive their abilities, and adopt behaviours in a socially and cognitively mediated environment. By empirically linking confidence calibration and perceived opinion leadership within a general population sample, this study contributes to communication research by demonstrating that communicative influence can emerge independently of domain-specific knowledge accuracy. The TSFCM provides a concept for understanding how individuals learn within a social network, while the DKE highlights potential barriers to accurate knowledge acquisition and behavioural adaptation within these networks. Despite their perceived authority, the knowledge and efficacy of opinion leaders can be hindered by cognitive biases such as the DKE. Thus, including both concepts in SCT can help explain why some individuals fail to adopt behaviours aligned with accurate health information (e.g., nutrition practices).

Accordingly, SCT and TSFCM are applied as guiding frameworks rather than comprehensive explanatory models, acknowledging that additional mechanisms such as motivated reasoning and identity-related heuristics may shape opinion leadership in contemporary communication environments. Against this background, the present study examines the role of opinion leaders within the German population with respect to agricultural knowledge, self-confidence, and media use.

Using a representative sample, it investigates whether perceived opinion leadership is associated with superior factual knowledge or whether confidence and communicative engagement play a more decisive role. In addition, the study explores how confidence calibration and cognitive biases relate to opinion leadership in the context of agricultural issues. Conceptually, perceived opinion leadership is treated as the outcome variable, while knowledge accuracy, self-confidence, media use, and selected sociodemographic characteristics are examined as associated explanatory factors.

## 2. Materials and methods

### 2.1. Sampling

The subject of this study was to survey the German society as a whole with regard to agricultural knowledge, media use on agricultural topics, and potential opinion leadership on these themes. A nationwide online survey was conducted among the German population aged 18 years and older between mid-September and mid-October 2023. Participant recruitment was facilitated through a field service provider using a nationwide consumer panel, which ensured the separation of personal and content-related data to maintain compliance with the European GDPR principles, ensuring secure and ethical data processing principles. Thus, personal identifying data were handled exclusively by the panel provider and were never accessible to the researchers. No questions enabling identification of individual participants were included in the questionnaire, and the study adhered to established ethical guidelines for research involving human participants, including informed consent. The panel allowed for pre-stratification of the sample to ensure representativeness of the German population across key demographic variables, including age, gender, residential area size, and federal state.

Although quota-based stratification ensured demographic representativeness regarding age, gender, and region, online panels may overrepresent digitally engaged individuals or those accustomed to survey participation. This methodological constraint is typical in large-scale public-opinion surveys and is addressed in the limitations section (see Wood et al. 2017 for general guidance on online panel data quality) [41].

The survey collected a range of sociodemographic information alongside data on leisure activities in rural areas, personal connections to agriculture, attitudes toward technology, local food production, sustainable consumption, and agricultural knowledge. To ensure data plausibility, multiple validation indicators for checking for internal consistency, analysing response times, and verifying logical dependencies between answers were applied [41]. To describe the data-validation protocol more explicitly, responses with implausibly short completion times (below one third of the median duration) were flagged for inspection, and patterns of straight-lining or contradictory responses in logically dependent items were checked. No automated or bot-like response patterns were detected. These procedures align with established survey-quality recommendations [41].

Following validation, the final dataset comprised 2,022 completed and valid responses. The distribution of the characteristics of the final overall sample proved to be suitable for representing the German population aged 18 and over in several characteristics (Table 1).

The characteristics of gender, age, and residential area size pre-stratified in the data collection are almost identical to the German official distributions [36], even after the final data check. The sample shows slightly more older people over the age of 60, while younger people under the age of 30 were slightly underrepresented. However, the remaining shares of age categories, as well as the distribution by residential area size (rural, urban), are well met. In terms of disposable household income, the highest category (more than 5,000 €) is significantly lower, while the mid-income categories match the distribution in the overall population. There are no official statistics on the respondents' relationship to agriculture in the form of their own professional experience or their immediate circle of acquaintances, but these characteristics are seen as relevant for categorizing the respondents in terms of their knowledge and opinions on agricultural topics [43,44]. Since only a small proportion of the sample (8.9%) has a professional relationship with agriculture and only one in five participants has a farmer in their immediate circle of friends, it would be expected that this minority would also have a better knowledge of agricultural concerns. The geographical distribution of the sample across the 16 federal states was

**Table 1. Distribution of sample characteristics (n = 2,022).**

| Characteristics | | abs. | rel. % | Germany[a] |
|---|---|---|---|---|
| Gender* | Female | 1.046 | 51.7 | 50.6 |
| | Male | 968 | 47.9 | 49.4 |
| | Non-binary | 8 | 0.4 | n/a |
| Age (in years)* | 18–30 | 343 | 17.0 | 21.2 |
| | 31–40 | 348 | 17.2 | 17.7 |
| | 41–50 | 329 | 16.3 | 16.2 |
| | 51–60 | 431 | 21.3 | 20.7 |
| | >60 | 571 | 28.2 | 24.3 |
| Disposable household income | <1.000 € | 230 | 11.4 | 8.6 |
| | 1.001–2.000 € | 537 | 26.6 | 24.7 |
| | 2.001–3.000 € | 465 | 23.0 | 23.5 |
| | 3.001–4.000 € | 370 | 18.3 | 16.2 |
| | 4.001–5.000 € | 219 | 10.8 | 10.8 |
| | >5.000 € | 201 | 9.9 | 16.1 |
| Highest educational qualification | Secondary school leaving certificate or no general qualification (yet) | 273 | 19.9 | 22.2 |
| | Intermediate school leaving certificate | 598 | 43.5 | 43.1 |
| | A-levels ("Abitur") | 503 | 36.6 | 34.7 |
| | University degree/ vocational training[2] | 648 | | n/a |
| Residential area size (number of inhabitants)[b] | <5.000 | 280 | 13.8 | 13.0 |
| | 5.001–20.000 | 511 | 25.3 | 24.9 |
| | 20.001–100.000 | 560 | 27.7 | 27.3 |
| | >100.000 | 671 | 33.2 | 34.8 |
| Relation to agriculture | Works in agriculture or related field | 180 | 8.9 | n/a |
| | Farmer in immediate circle of friends | 385 | 19.0 | n/a |

Note: [a]Source: [42]; Comparison with official national statistics not possible. only applicable to the first three educational qualification categories.
[b]Pre-stratificated sample demographic variables.

representative due to the pre-stratification method and is not shown in Table 1. Since the sampling already followed quota-based representativeness and deviations remained minor (see Table 1), no additional post-stratification weighting was performed.

## 2.2. Analysis methods

The testing and analysis of the survey data was carried out with MS Excel and the statistical software SPSS 28.0. Several partly multivariate analysis methods were used for the analysis. Opinion leadership (OLI) serves as the dependent variable in subsequent regression models, while independent variables include knowledge accuracy, mean confidence, over-/ underconfidence metrics, media-use factors, sociodemographic variables, and agricultural background indicators.

**2.2.1. Measuring knowledge and self-confidence.** The online survey collected information on the German population's views on current agricultural practices, with a focus on measuring explicit agricultural knowledge. To assess this, the approach outlined by [45] was adopted, incorporating nine knowledge questions presented in a binary true/false format. This empirical methodology builds on prior studies (e.g., [46,47]), which examined public perception, knowledge levels, and confidence regarding topics such as climate change and natural phenomena.

For the context of German agriculture, several statements were developed by experts from various agricultural sectors and pre-tested for clarity and validity. Experts were instructed to formulate statements that an average, actively interested

member of the general population would be able to evaluate. A set of ten initial statements (later reduced to nine) was formulated and validated by subject-matter experts from crop science, agricultural economics, livestock science, and environmental science. Their task was to ensure factual correctness, clarity, and alignment with issues that a generally interested public encounters. The questionnaire originally asked a total of ten statements. Subsequently, one question had to be removed because it was imprecisely formulated and therefore the respondents' answers could not be clearly determined. Of the nine remaining statements, four were true, while the remaining five were deliberately formulated in a false manner. This balanced distribution was designed to account for the "veracity effect," a cognitive bias where individuals are more likely to perceive statements as true rather than false [48:125].

In the words of the authors, "truths are most often correctly identified as honest, but errors predominate when lies are judged." By maintaining equal representation of true and false formulated statements, this design facilitates an accurate assessment of respondents' knowledge, including potential overestimations and underestimations. Linking this design to relevant cognitive-bias literature, the veracity effect has been highlighted as a robust bias in judgment research (e.g., Deceptive Communication Theory [49], and memory-based truth judgments [50]). Thus, the deliberate inclusion of both, true and false statements reduces systematic bias in accuracy measurement. Respondents were also asked to indicate their confidence in their answers using a six-point subjective probability scale. Confidence levels ranged from 50% (indicating a pure guess with a 50/50 chance) to 100% (indicating absolute certainty), increasing in 10% increments. This scale follows the framework established by [51]. It should be noted that discrete probability scales correspond to classical calibration research [52], which recommends evenly spaced probability steps to detect systematic over-/underconfidence. Therefore, the scale is consistent with confidence elicitation standards in judgment and decision-making research. By combining these two data points—objective knowledge accuracy and subjective confidence levels—it was possible to calculate correct rates and confidence metrics at three levels: overall, individual respondent, and individual question. Thus, this dual analysis provides a comprehensive understanding of knowledge accuracy and confidence dynamics in the context of German agriculture. Table 2 illustrates the nine statements and the answer options as presented to the

**Table 2. Queried statements and confidence levels.**

For the following nine statements, indicate whether they are correct or incorrect (left-hand side). Also, indicate how convinced you are of your judgement (right-hand side)

| Statement (with actual correct answer) | True | False | 50% | 60% | 70% | 80% | 90% | 100% |
|---|---|---|---|---|---|---|---|---|
| | | | (= I have to guess) | | | | (= I know that for a fact) | |
| 1. Self-sufficiency in pork in Germany is below 100%. (no) | ☐ | ☐ | ☐ | ☐ | ☐ | ☐ | ☐ | ☐ |
| 2. In agriculture in Germany, grain cultivation is of the greatest importance. (yes) | ☐ | ☐ | ☐ | ☐ | ☐ | ☐ | ☐ | ☐ |
| 3. A farmer in Germany can nowadays feed more than 1,000 people. (no) | ☐ | ☐ | ☐ | ☐ | ☐ | ☐ | ☐ | ☐ |
| 4. In Germany, about one-third of all wild animal species are classified as "strongly endangered" or "endangered". (yes) | ☐ | ☐ | ☐ | ☐ | ☐ | ☐ | ☐ | ☐ |
| 5. The cultivation of grass and clover reduces soil erosion and soil loss from the fields. (yes) | ☐ | ☐ | ☐ | ☐ | ☐ | ☐ | ☐ | ☐ |
| 6. The number of heavy rainfall events in Germany has been decreasing for years. (no) | ☐ | ☐ | ☐ | ☐ | ☐ | ☐ | ☐ | ☐ |
| 7. Sandy soils are good reservoirs for water and plant nutrients. (no) | ☐ | ☐ | ☐ | ☐ | ☐ | ☐ | ☐ | ☐ |
| 8. About 30% of Germany's land area is used for agriculture. (yes) | ☐ | ☐ | ☐ | ☐ | ☐ | ☐ | ☐ | ☐ |
| 9. In organic farming, synthetic chemical pesticides can be used to a certain extent. (no) | ☐ | ☐ | ☐ | ☐ | ☐ | ☐ | ☐ | ☐ |

respondents in the online survey. To exclude bias due to sequence effects, the nine statements were shown to the respondents in random order. At the statement level, the mean correct rate across all respondents was determined for each of the nine statements. This can also be done for the sum of all nine statements and for the sum of the four correct statements and the five incorrect statements. The mean correct rate per question was also decisive for categorizing the nine questions into three *ex-post* assigned statement accuracy level 1, 2, and 3 (SAL).

SAL can be used to show which statements were generally more difficult to answer (with a lower mean correct rate) than others. The self-assessed confidence scale enabled the calculation of mean confidence levels for each statement, as well as aggregated means across all nine statements and separately for the four true and five false statements. The number of responses indicating 50% confidence for a given statement was used to calculate a "guess rate" (GR), while the number of responses indicating 100% confidence provided a "knowing rate" (KR). To further evaluate the relationship between knowledge and confidence, the mean confidence was subtracted from the mean accuracy (correct response rate) for each statement, yielding a percentage value referred to as "Over-Underconfidence" (OU). This metric captures discrepancies between confidence and actual knowledge and can take on both, positive and negative values. Positive OU values indicate overestimation of knowledge, negative values signify underestimation, and values near zero reflect an appropriate alignment between confidence and knowledge ("good confidence fit"). This analytical approach enables a nuanced assessment of knowledge and confidence in agricultural topics, both at the level of individual statements and at an aggregated level, offering insights into respondents' understanding and their self-perception of expertise. In addition, it is possible to estimate the Dunning-Kruger effect. At the individual level, the equation proposed by Jonsson and Allwood [53] can be applied to evaluate whether an individual is overconfident, underconfident, or acceptably calibrated in terms of their knowledge and self-confidence.

$$\frac{1}{n} \sum_{i=1}^{T} n_t (r_{tm} - c_t)$$

(1)

In this equation, *n* represents the total number of statements used in the survey ($n = 9$), and *T* denotes the number of confidence levels ($T = 6$). The variable $n_t$ represents the frequency with which each confidence level is used, while $c_t$ is the proportion of correct answers corresponding to each confidence level. $c_t$ is then subtracted from the mean confidence rating for each confidence level, $r_{tm}$. For individual respondents, a negative value resulting from this equation indicates underconfidence, whereas a positive value signifies overconfidence. Individuals whose calculated values fall between −5% and +5% are classified as acceptably calibrated, with those achieving exactly 0% labelled as "well-calibrated." By examining the extent and distribution of deviations at the individual level, this approach also allows for an assessment of overconfidence and underconfidence trends across the overall sample. To meet the requirements for a clear interpretation of this formula and its grounding in judgment calibration literature requirement, it should be noted that this approach stems from classical overconfidence research (e.g., [54–56]), where discrepancies between subjective probability and empirical correctness represent miscalibration. The −5% to +5% interval corresponds to thresholds used in prior studies applying Jonsson & Allwood's metric. Furthermore, the individual confidence values obtained through this analysis can be utilized for subsequent investigations, such as exploring interactions with opinion leadership, as described in the following.

**2.2.2. How to assess opinion leadership.** The status of opinion leadership was measured and assessed using the revised and established opinion leadership scale developed by Childers [55]. The scale had been further refined by Childers from earlier measurements (e.g., [57]), and its validity was tested in relation to variables such as consumers' risk preferences and ownership experience. For this study, the original seven-item scale was adapted to suit the context of agricultural knowledge, and its internal reliability was tested in advance. A total of five five-point Likert-scale items were used, addressing the dominant direction of communication with peers, as well as the extent and frequency of opinion sharing (Table 3). It should be explicitly highlighted that the adapted Childers scale primarily captures self-perceived communicative influence rather than actual behavioural influence. This aligns with previous applications of

**Table 3. Used scales to measure and index, opinion leadership (OLI).**

| Items/answer categories | 1 | 2 | 3 | 4 | 5 |
|---|---|---|---|---|---|
| 1. When I talk to acquaintances or friends about agriculture and food. I... Information. | … bring in a great deal of … | ... bring in some... | … don't provide much … | … bring in almost no … | … don't bring in any … |
| 2. How many people in your circle of acquaintances have been informed by you about agriculture and food in the last six months? | Very many people | Several people | Few people | Almost no people | Nobody |
| 3. If you (would) talk to acquaintances or friends about agriculture and food, which situation is more likely? | I set the tone completely on these topics. | I set the tone a little more. | Con-versations are balanced. | My friends set the tone a little more. | My friends are fully in charge of this topic. |
| 4. How often are you consulted for advice on agriculture and food in discussions with your friends and acquaintances? | Regularly | Frequently | Occasionally | Rarely | Never |
| 5. How often do you talk about agriculture and food with your friends and acquaintances? | Regularly | Frequently | Occasionally | Rarely | Never |
| **Used index value (IV)** | **+2** | **+1** | **0** | **−1** | **−2** |

opinion-leadership metrics in consumer and agricultural communication research (e.g., adaptation studies in food-risk communication). Furthermore, it should be clarified that the Childers scale overlaps with related constructs such as "market mavenship" and "influencer communicativeness". While these constructs are not measured in this study, the OLI may reflect primarily communicative confidence and willingness to share information rather than domain expertise alone. It should therefore be emphasized that the OLI captures self-perceived communicative influence rather than objectively verified behavioural influence or network centrality.

To facilitate analysis, the responses were converted into index values (IV) ranging from +2 to −2. These index values were aggregated and averaged to calculate an "opinion leadership index" (OLI).

$$OLI = \sum_{q=1}^{n} (IV)/n$$

(2)

Through this process, a metric was generated for each individual in the sample, indicating the extent to which they functioned as an opinion leader or as an opinion receiver. Internal consistency (Cronbach's α) was recalculated and documented in the results chapter. The scale demonstrated satisfactory reliability, supporting its applicability in the agricultural-knowledge context.

**2.2.3. Media use.** The two-step communication model introduced by Katz and Lazarsfeld was utilized [34], which posits that opinion leaders are first influenced by (mass) media on a given topic before communicating information to their social environment. To capture the extent and type of media use, respondents were asked about the frequency with which various media formats were used to obtain information on agricultural topics and food. Six media formats were selected, encompassing both online media (e.g., social media, internet forums, and blogs) and offline media (e.g., radio and television), as well as specialized sources (e.g., trade journals, producer websites) and general sources (e.g., daily newspapers). The selection of media format categories was derived and adapted from [58]. The frequency of use for each format was recorded using a five-point scale ranging from 'regularly' to 'never.'

The data obtained were analysed using a principal component matrix with varimax rotation (total variance explained: 70.2%, KMO 0.777, $p = 0.001$). This analysis allowed the six media formats to be classified into two factor components: 'classic offline media' and 'digital online media.' Individual factor values were assigned to each survey participant for both, components, indicating the intensity with which these two media sources were utilized. These individual factor values for both components can be used for further analyses to investigate relationships with other factors. Component extraction was based on the Kaiser criterion (eigenvalues > 1), and both components showed clear interpretability, with

strong loadings (> 0.60) on theoretically coherent item groups. As link to the Multi-Step Flow of Communication, the separation between offline and online channels reflects different informational pathways: offline media typically align with the classical two-step flow, whereas digital media facilitate multi-step, peer-to-peer diffusion processes.

**2.2.4. Analysing interrelations.** Several multivariate methods were used to analyse the interactions between respondents' correct knowledge rates, their self-confidence, their tendency to overestimate themselves, and their status as opinion leaders. One analysis step employs a correlation matrix, calculating Pearson correlation coefficients to measure the strength and direction of relationships between variables. Key indicators include the correct knowledge rate, mean confidence across all statements, over-to-underconfidence measures (as per the approach of [45]), and the Opinion Leadership Index (OLI). These correlations highlight how knowledge and confidence interact with opinion leadership tendencies. Pearson's r was chosen because the variables involved (accuracy scores, confidence means, OLI, and factor scores) were continuous and approximately normally distributed, fulfilling the basic assumptions for parametric correlation analysis. In a further step, a comparison of the top and bottom quintiles of the OLI distribution was conducted to identify key differences between opinion leaders and (strong) opinion receivers in the last quintile of the sample. The approach involves mean value comparisons for variables such as media usage (offline and digital), confidence levels, overconfidence tendencies, and contextual self-assessment abilities. Statistical measures include significance testing (t-tests) and effect size calculations using Cohen's d to quantify the magnitude of observed differences and to provide insight into how opinion leaders differ from opinion receivers in behaviour, media engagement, and cognitive traits. Comparisons focusing on groups at the ends of the distribution are commonly used in opinion-leadership research, as they allow clearer contrasts between individuals with pronounced leadership tendencies and those with pronounced opinion-receiving tendencies. However, this approach emphasizes differences at the distribution's ends and does not describe patterns across the entire spectrum.

In the final step, a linear regression model is used to predict the OLI as a dependent variable based on various explanatory factors, such as knowledge accuracy, mean confidence, media usage (offline and online), sociodemographic variables, and personal connections to agriculture. The model calculates unstandardized (B) and standardized (Beta) coefficients to assess the relative influence of predictors, along with statistical significance and confidence intervals for precision. This analysis identifies key characteristics of opinion leaders and the extent to which confidence, media engagement, and personal connections shape their role. All standard regression diagnostics were conducted to confirm that the regression model meets the required assumptions:

- Variance Inflation Factors (VIFs) were well below the critical value of 5, indicating no problematic multicollinearity.

- Visual inspection of residual plots confirmed homoscedasticity.

- Normality of residuals was assessed via Q-Q plots, indicating acceptable distribution.

- Cook's distance values were all below 1, suggesting no influential outliers.

Additionally, an interaction term (knowledge accuracy × mean confidence) was tested to examine potential moderation effects. The interaction was not statistically significant ($p > 0.05$), indicating that the relationship between knowledge accuracy and opinion leadership does not depend on confidence levels. The interpretation of the regression results has been aligned with the central finding that self-confidence and media use—rather than factual knowledge accuracy—are stronger predictors of opinion leadership within the general population.

## 3. Results and discussion

This chapter presents the findings of the study, exploring the relationships between agricultural knowledge, self-confidence, media use, and opinion leadership, followed by an in-depth discussion of their implications in the context of public understanding and communication dynamics.

## 3.1. Knowledge and self-confidence

How well do people really understand agricultural issues, and does their confidence match their knowledge, or are there cognitive biases that tip the balance? In the following, patterns of accuracy, confidence calibration, and the influence of cognitive biases such as the Dunning-Kruger effect are identified by analysing respondents' agricultural knowledge and confidence. To address the study objectives, this section first reports descriptive patterns of knowledge and confidence and then links these patterns to indicators of miscalibration (over- and underconfidence) at both, item and individual level.

### 3.1.1. Knowledge about agriculture in Germany.

The accuracy of the responses to the four true and five false statements on agricultural topics was determined on the basis of the survey of 2,022 participants (Table 4). The overall mean correct response rate across all nine statements is 64%, with individual statement accuracy ranging from 41% to 78%. Standard deviation at the individual statement level is correspondingly high, while that across all nine questions is reduced. Statements such as "The number of heavy rainfall events in Germany has been decreasing for years." achieved the highest correct rate (78%), indicating greater public familiarity with the fact that high-rain events have increased, while statements like "Self-sufficiency in pork in Germany is below 100%" had the lowest correct rate (43%), exemplary reflecting knowledge gaps.

According to the mean correct rate, statements were *ex post* categorised into three statement accuracy levels based on their correct response rates. SAL1 (high) includes three statements with mean correct rates above 70%, such as "The cultivation of grass and clover reduces soil erosion and soil loss from the fields." SAL2 (medium) comprises three statements with mean correct rates between 60% and 70%, such as "In agriculture in Germany, grain cultivation is of the greatest importance." Level 3 (low) includes three statements with correct rates below 60%, such as "A farmer in Germany can feed nowadays more than 1,000 people," which was the most difficult statement to assess with the lowest accuracy level. 95% confidence intervals provide additional reliability for the correct rates. These results highlight substantial differences in public knowledge of agricultural topics. Some areas, such as emerging threats to wildlife species or the increasing

**Table 4. Measured knowledge on statement level.**

| Statements[b] | T/F | Mean correct rate | SD | CI[a] | | |
|---|---|---|---|---|---|---|
| | | | | 95/- | 95/+ | SAL |
| 1. Self-sufficiency in pork in Germany is below 100%. | No | .43 | .495 | .40 | .45 | 3 |
| 2. In agriculture in Germany, grain cultivation is of the greatest importance. | Yes | .67 | .471 | .65 | .69 | 2 |
| 3. A farmer in Germany can nowadays feed more than 1,000 people. | No | .41 | .493 | .39 | .44 | 3 |
| 4. In Germany, about one third of all wild animal species are classified as "strongly endangered" or "endangered". | Yes | .76 | .429 | .74 | .78 | 1 |
| 5. The cultivation of grass and clover reduces soil erosion and soil loss from the fields. | Yes | .77 | .424 | .75 | .78 | 1 |
| 6. The number of heavy rainfall events in Germany has been decreasing for years. | No | .78 | .413 | .76 | .80 | 1 |
| 7. Sandy soils are good reservoirs for water and plant nutrients. | No | .70 | .459 | .68 | .72 | 2 |
| 8. About 30% of Germany's land area is used for agriculture. | Yes | .66 | .472 | .64 | .68 | 2 |
| 9. In organic farming, synthetic chemical pesticides can be used to a certain extent. | No | .56 | .496 | .54 | .58 | 3 |
| Total | | .64 | .148 | .63 | .64 | |

Note: T/F = True (yes) or False statement (no); SD = Standard deviation; SAL = Ex-post assigned statement accuracy level according to share of correct answers: 1) >0.70; 2) >0.60 and <= 0.70; 3) <=0.60; [a]95% confidence intervals using bootstrapping with 1,000 resamples. [b]statements were shown to the respondents in random order.

importance of soil cover to prevent erosion, are relatively well understood. However, significant misconceptions or gaps in knowledge exist, particularly concerning the performance of domestic producers in the provision of food and detailed legal restrictions on organic production. It should be noted that the nine items cover a limited set of salient agricultural topics and cannot capture agricultural knowledge in its full breadth. The findings therefore reflect performance on these specific statements rather than a comprehensive literacy index.

The overall mean correct response rate consists of the four correctly formulated statements (Statements 2, 4, 5, and 8), which achieved a higher mean correct rate of 0.70, and the five incorrectly formulated statements (Statements 1, 3, 6, 7, and 9), which had a lower mean correct rate of 0.58. This discrepancy reflects the veracity effect, as described in the literature, whereby errors are more likely to occur when evaluating incorrectly formulated statements [43]. Comparable findings have been reported in similar studies. For instance, an Austrian study by Thaller and Brudermann [39] observed a comparable difference between the "hit rate" for true statements (60%) and the "correct rejection rate" for false statements (51%). A more pronounced methodological bias was identified in the study by Fischer et al. [40], which assessed knowledge of climate change, where the "hit rate" and "correct rejection rate" differed more substantially at 73% and 48%, respectively. These results highlight the influence of statement formulation on response accuracy and the persistence of veracity effects across different domains of knowledge assessment. It can therefore also be deduced that deliberately incorrectly formulated statements or news are more frequently accepted as 'true'. Taken together, the pattern is consistent with previous work on truth bias and veracity effects, underlining that response accuracy is not only a function of knowledge but also of item formulation and cognitive heuristics in judging statements as true or false.

**3.1.2. Self-confidence.** Mean confidence rates and shares of over- or under-confidence (OU) were assessed to determine whether participants were overestimating or underestimating their abilities overall and on individual statement levels (Table 5). Overall, participants demonstrated a mean confidence of 0.70 across all statements, with a 'guess rate' (GR) of 0.31 and an 'absolute certainty rate' (AC) of 0.13. The AC/GR ratio was 0.51, reflecting moderate certainty in all responses. However, participants exhibited a slight tendency toward overconfidence, as indicated by an OU of −6%. This suggests that, on average, participants believed they were more accurate than they actually were. By construction of the OU measure (accuracy minus confidence), negative values indicate overconfidence and positive values indicate underconfidence. The observed −6% thus reflects modest but systematic overestimation of one's own accuracy at the aggregate level.

**Table 5. Rates of mean confidences and assessment of over- and underconfidence.**

| SAL | Statements (T/F) | Mean Confidence | GR | AC | AC/GR | OU-% | OU Ass. |
|---|---|---|---|---|---|---|---|
| 1 | 6. The number of heavy rainfall events … (no) | .80 | .14 | .28 | 1.95 | −2% | Good |
| | 5. The cultivation of grass and clover …(yes) | .70 | .34 | .13 | .38 | +7% | Under |
| | 4. In Germany, about one third of all wild … (yes) | .70 | .27 | .08 | .29 | +6% | Under |
| 2 | 7. Sandy soils are good reservoirs for water … (no) | .73 | .28 | .18 | .65 | −3% | Good |
| | 2. In agriculture in Germany, grain … (yes) | .68 | .31 | .08 | .24 | +1% | Good |
| | 8. About 30% of Germany's land area is … (yes) | .64 | .42 | .05 | .12 | +2% | Good |
| 3 | 9. In organic farming, synthetic chemical... (no) | .72 | .28 | .14 | .49 | −16% | Over |
| | 1. Self-sufficiency in pork in Germany … (no) | .68 | .39 | .13 | .32 | −26% | Over |
| | 3. A farmer in Germany can nowadays feed... (no) | .66 | .41 | .08 | .19 | −24% | Over |
| **Total (all statements)** | | **.70** | **.31** | **.13** | **.51** | **−6%** | **Over** |
| True statements (yes) | | .68 | .33 | .08 | .26 | +3% | Good |
| False statements (no) | | .72 | .30 | .16 | .72 | −14% | Over |

Note: T/F = True (yes) or False (no); GR = 'Guess rate' – stated 50% confidence; AC = 'absolute certainty' – stated 100% confidence; SAL = Assigned statement accuracy level according to share of correct answers: 1) >0.70; 2) >0.60 and <= 0.70; 3) <=0.60; OU = "Over-under-confidence" = "mean correct rate" – "mean confidence"; OU Assessment = 'Underestimation' >= 5%; 'Overestimation' <= −5%; Good −5% to 5%; (n = 2,022 Participants)

When comparing performance between true (yes) and false (no) statements, notable differences emerged. For true statements, the mean confidence was 0.68, and participants displayed good calibration with an OU% of 3%, suggesting they were neither pronounced over- nor under-confident. Conversely, false statements had a higher mean confidence of 0.72 but showed significant overconfidence with an OU% of −14%. This disparity indicates that, similar to the mean correct rate, participants struggled more with false-formulated statements, often overestimating their understanding.

Accuracy level also influenced confidence performance. For the statements with the highest accuracy level (SAL1), mean confidence was high, ranging from 0.70 to 0.80, with relatively low GR and favourable AC/GR ratios. The OU for these statements ranged from −2% to 7%, with assessments categorised as either 'underconfidence' or 'good,' indicating well-calibrated confidence for straightforward content. For medium accuracy level (SAL2), mean confidence decreased slightly to 0.64–0.73. The three statements on this level were assessed as 'good' in terms of OU. For the low accuracy statements (SAL3), confidence ranged from 0.66 to 0.72, but participants exhibited high GR and overconfidence. The OU ranged from −16% to −26%, highlighting a consistent tendency to overestimate their accuracy for more difficult content. For example, responses to statement 3 (OU: −26%) and statement 1 (OU: −24%) illustrate marked overestimation of knowledge for false and difficult-to-answer statements. These results suggest cognitive biases or gaps in critical evaluation, particularly when assessing incorrect information. In contrast, some true statements showed underconfidence (e.g., statements 4, 5, and 8), indicating that participants were more successful at correctly evaluating true information, even at medium statement accuracy levels. This asymmetry between true and false statements complements the veracity-effect pattern reported above, suggesting that people are relatively cautious when confronted with true statements but more prone to overconfidence when accepting false ones.

Self-confidence was tested in six 10%-steps from 50% to 100%. The distribution patterns of the self-assessment percentages of the nine statements differed to some extent (Fig 2). In most cases, a large proportion of respondents stated that they totally guessed the answer (= 'guess rate', around 31% on average of all statements). After that, proportions increase before fewer respondents become highly certain about the correctness of their answers. These tendencies are not necessarily dependent on the accuracy level of a statement. A closer look at the statements with SAL1, the progression pattern, and the proportions of people who are absolutely certain clearly differ (solid lines for statements 4, 5, & 6).

Overall, the distributions suggest that many respondents adopt a cautious stance (high GR) but that a non-negligible subgroup reports very high certainty even for difficult items, which is consistent with the miscalibration patterns reported above. The survey data also allow over- and underconfidence to be measured at the individual level. The method proposed by Jonsson and Allwood [53] quantifies individual confidence calibration by evaluating over- or underconfidence across the set of nine statements. This approach incorporates both, the distribution of confidence levels and the proportion of correct answers at each level. Negative values indicate underconfidence, while positive values denote overconfidence, categorised into detailed ranges like '>50%', '+20% to +50%', and '+5% to +20%' for overconfidence, and similar ranges for underconfidence (Fig 3). Values of 0% reflect 'well-calibrated' individuals, and those within ±5% are 'acceptably calibrated.'

More than half of the participants (51.7%) are overconfident, while significant underconfidence is seen in 21.8% (5% to −20%) and 6.9% (−20% to −50%). Extreme cases of overconfidence (>50%) and underconfidence (≤50%) are rare, at 1.2% and 0%. Only 0.9% are 'well-calibrated,' but the broader −5% to +5% range includes nearly one fifth of all participants. However, most (71.2%) fall within '-20% to +20%', reflecting reasonable calibration overall, though an average score of +6.3% (SD = 0.19) indicates a slight tendency toward overconfidence. Thus, while a majority shows some degree of over- or underconfidence, severe miscalibration is relatively rare, and most individuals cluster in moderate ranges around zero, which is in line with the literature on confidence calibration in general-knowledge tasks.

**3.1.3. Self-assessment and Dunning-Kruger effect.** Following their answers to the nine statements, survey participants were asked to evaluate how many questions they answered correctly compared to the German population (GP). This self-assessment aimed to contextualize their answers and confidence levels on a broader scale. Specifically,

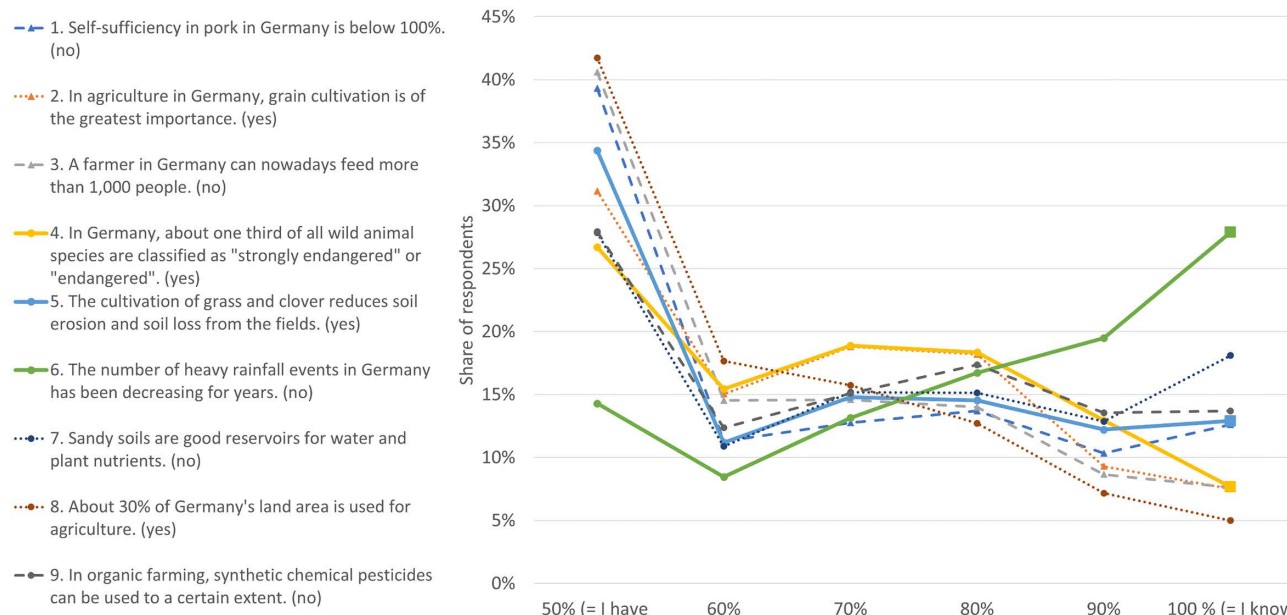

**Fig 2. Distribution patterns of the six confidence levels per statement.**

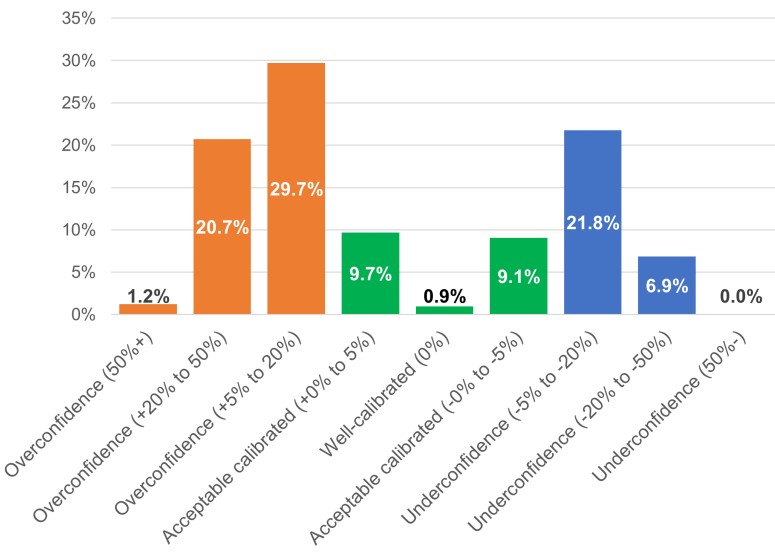

**Fig 3. Distribution of individuals to levels of over- and underconfidence (n = 2,022).**

the 2,022 respondents were asked to indicate whether they believed they answered fewer questions correctly than 50% of the German population, or more than 50%, 75%, or even more than 90% of their fellow citizens (Table 6).

This result suggests a pattern that is compatible with the Dunning-Kruger discussion, as a majority of respondents appear to rate their performance above the statistical average, indicating a general tendency toward overestimation. However, this pattern alone is not sufficient to establish the full Dunning-Kruger effect in the strict sense.

**Table 6. Comparison of self-assessment accuracy between confidence calibration categories.**

| Self-assessment compared to the German population (GP) | Distribution in % | | | |
|---|---|---|---|---|
| | Total sample | Under-confident | Acceptable (incl. well-) calibrated | Over-confident |
| N | 2,022 | 579 | 398 | 1,045 |
| I got fewer answers correct than 50% of the GP | 29.6 | 48.0 | 29.9 | 19.3 |
| I got more answers correct than 50% of the GP | 50.2 | 44.6 | 57.3 | 50.6 |
| I got more answers correct than 75% of the GP | 17.3 | 6.6 | 12.8 | 24.9 |
| I got more answers correct than 90% of the GP | 2.9 | .9 | .0 | 5.2 |

Subsample crosstab: Pearson-Chi²: $\chi^2$ = 227.387, df=6, p<0.001.

This result suggests possible evidence of the Dunning-Kruger effect, as the majority of respondents overestimate their abilities, reflecting a general cognitive bias in which people with limited knowledge do not recognise their own incompetence [20,59]. The proportion of those who think they are outperforming large parts of the rest of the population then decreases continuously. Only a small fraction (2.9%) viewed their performance as exceptional, surpassing 90% of the population. A significant proportion of under-confident respondents (48.0%) believed they answered fewer questions correctly than 50% of the population. Only less than 1% believed they exceeded 90% of the population. Over-confident respondents frequently overestimated their performance, with 50.6% believing they performed better than 50% of the population, 24.9% claiming to surpass 75%, and 5.2% estimating their performance exceeded 90% of their peers. A Pearson Chi-squared test reveals a significant association between the three confidence calibration categories and self-assessed performance. Overall, under-confidence is marked by significant underestimation of one's performance, while over-confidence involves a substantial overestimation of ability compared to peers. Acceptable calibration aligns more closely with realistic self-assessments. The significant $\chi^2$ result indicates these patterns are not due to random variation but reflect systematic differences in confidence calibration.

By assigning the confidence level with a consecutive correct knowledge rate, it is possible to check whether the results of the survey of the German population reflect the typical Dunning-Kruger progression curve which shows perceived competence versus actual competence (compare Fig 1). "Little knowers" often overestimate their abilities ("Peak of inflated confidence"). Intermediates recognize their limitations ("Valley of despair"). Experts slightly underestimate themselves but are accurate overall. Fig 4 illustrates the progression over all nine statements between correct knowledge rates of 22% and 89%. The solid line represents the average confidence values, while the light grey band denotes the 95% bootstrap confidence intervals across all nine statements.

The shape of the line reflects the early stages of the Dunning-Kruger curve. An initial peak in confidence is observed at a knowledge rate of around 44%. Beyond this point, confidence decreases slightly or stabilises before rising as correct knowledge rates increase. However, general conclusions regarding the significance and strength of this effect are challenging, as changes in the confidence rate occur within a relatively narrow range of 66% to 74%. The observed pattern should therefore be interpreted as an indication of structured miscalibration that resembles the early stages of the Dunning-Kruger curve rather than as definitive proof of the full canonical effect. The literature on the Dunning-Kruger Effect (DKE) suggests that the severity and complexity of the statements to be answered play a crucial role in shaping the effect. To analyse this, the statements were *ex-post* categorized into three statement accuracy levels: SAL1, SAL2, and SAL3. For each statement accuracy level, the mean confidence level and knowledge rate were calculated, and both, the mean values and curve progressions were statistically compared (see S1 Table). Third-degree polynomial functions were modelled to describe the curve progression for the aggregated data and each statement accuracy. Using an interaction model, the coefficients of the polynomial functions for the three accuracy levels were tested for statistical significance.

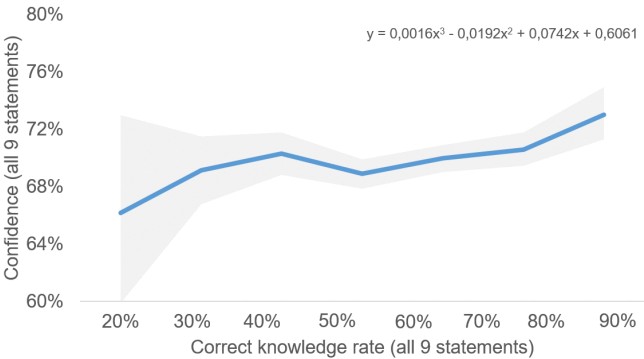

$$y = 0{,}0016x^3 - 0{,}0192x^2 + 0{,}0742x + 0{,}6061$$

**Fig 4. Interplay of correct knowledge rate and confidence levels over all statements (n = 2,022).**

The statistical comparison of the function coefficients, as well as the residual analysis (ANOVA, f-statistics), showed that the coefficients of all three levels vary significantly. SAL1 (high accuracy) shows the strongest deviation, while SAL2 and SAL3 (low) exhibit more moderate differences. Comparison of mean values via an ANOVA test show also significant differences ($F = 7.44$; $p = .0007$). SAL1 has the highest average confidence, while SAL3 systematically shows lower values. The curve progressions of the polynomial functions were compared statistically. SAL3 deviates substantially from the aggregated function, showing a more negative slope. SAL1 exhibits a more positive deviation, reflecting higher confidence levels with increasing correct answers. SAL2 resembles the aggregated function and shows no significant differences between the polynomial functions. These results suggest that the DKE is most pronounced in tasks of medium and high difficulty with a lower knowledge accuracy. The curve of the true statements starts lower and shows a steeper increase up to a comparable final value at 100% correct answers and differs significantly from the curve function of the aggregated model, for which the Dunning-Kruger effect is recognisable. The curve of the false statements is similar to the aggregated function. It is therefore shown that the true/false formulation and wording of a statement also have an influence on the magnitude of the effect. In sum, we find statistical patterns that are consistent with core ideas of the Dunning-Kruger literature (systematic miscalibration at lower accuracy levels and differences by item difficulty), but these results should be seen as evidence for structured miscalibration rather than as a full reconstruction of the canonical DKE curve.

### 3.2. Determination of opinion leadership and media use

Are people who are considered more knowledgeable or experienced on a topic also to be equated with opinion leaders? To calculate an individual Opinion Leadership Index (OLI), responses to six Likert-scale items were aggregated and averaged. The resulting index reflects whether a person tends to act as an opinion leader (OLI > 0.0) or is more influenced by others as an opinion receiver (OLI < 0.0), whereby a neutral position is assumed at the actual value of 0.0. Each respondent is assigned an OLI value ranging from −2 to +2. The distribution of these values, divided into 0.2 intervals, illustrates the overall allocation and the proportion of individuals with a positive OLI (Fig 5). Respondents with a positive OLI, indicating a higher likelihood of being opinion leaders, constitute 20.3% of the sample. In contrast, 71.8% of respondents have a negative OLI, categorizing them as rather opinion receivers. A smaller group, 7.9%, has a neutral index value (0.0). The mean OLI for all participants is −0.57 (SD = 0.773), emphasizing that the majority of the sample does not fulfil the role of opinion leader in the context of agricultural and food topics. For further analysis, the two extreme groups—opinion leaders (top quintile, Q1) and strong opinion receivers (bottom quintile, Q5)—are compared to investigate their interactions with actual knowledge and self-confidence in the following chapter 3.3. It is important to note that the OLI is based on self-reported communicative behaviour and perceived influence rather than sociometric network measures. The identified

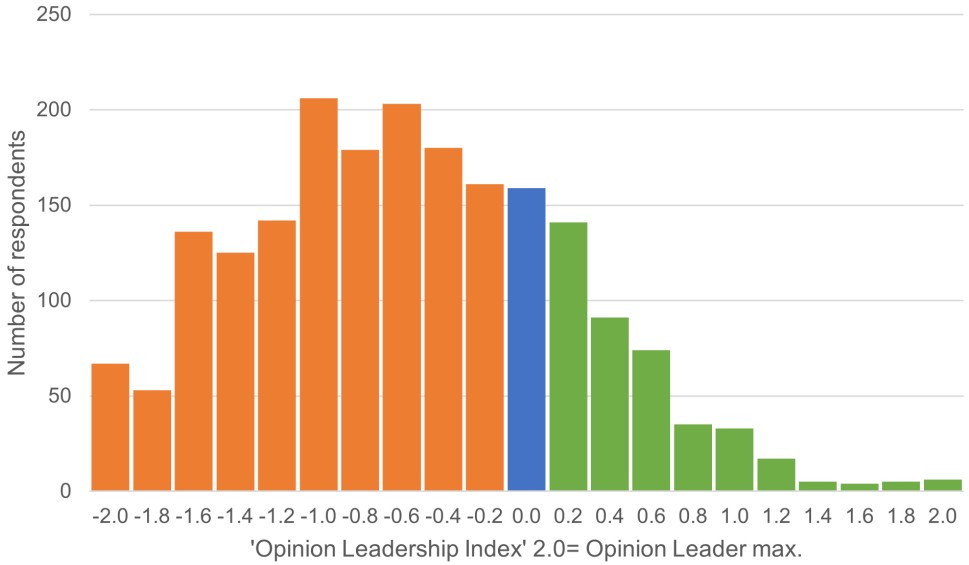

**Fig 5. Distribution of individual OLI values (n = 2,022).**

opinion leaders therefore represent individuals who see themselves as influential in their social circles, which is consistent with the operationalisation used in many opinion-leadership studies.

Survey participants were further asked about their use of media to gather information on agricultural topics and food (Fig 6). Traditional media, such as radio, television, internet sources, and daily newspapers, are used regularly by 18%, 12%, and 10% of respondents, respectively. However, a substantial proportion rarely or never uses these sources, with 37% for radio/TV and 61% for newspapers. Internet-based platforms, including blogs, forums, and podcasts, are moderately used, with 10% of respondents accessing them regularly, though 50% rarely or never do. Social media, like Facebook and X, is underutilized for these topics, with 43% never engaging. Specialized channels, such as producer websites or trade journals, have the lowest reach; only 8% and 6% use them regularly, while around half never consult them. The results highlight a divide in media consumption habits. While some respondents regularly use a range of sources, many remain disengaged, particularly with specialized and digital platforms. This suggests differences in interest, trust, or accessibility, offering opportunities to refine communication strategies to better reach less-engaged audiences. A principal component analysis groups these media formats into two separate media groups: 'classic offline media' and 'digital online media,' reflecting distinct patterns of media consumption. Possible differences in the use of these two groups are analysed in the following chapter. The grouping into offline and online media is consistent with the communication-theoretical distinction between traditional mass-media pathways and more interactive, multi-step digital channels. In the next section, these media-use factors are linked to opinion leadership to assess whether more engaged media users are also more likely to perceive themselves as opinion leaders.

### 3.3. Exploring the dynamics of opinion leadership

Multivariate analyses explore the interrelations between knowledge, confidence, and opinion leadership and characterise opinion leaders in contrast to the general population.

**3.3.1. Relationships between knowledge, confidence, and opinion leadership.** One of the research questions of this study was whether opinion leaders on agricultural and food topics also have a higher level of knowledge and self-confidence regarding their knowledge. A correlation matrix examines the relationships between correct knowledge rate,

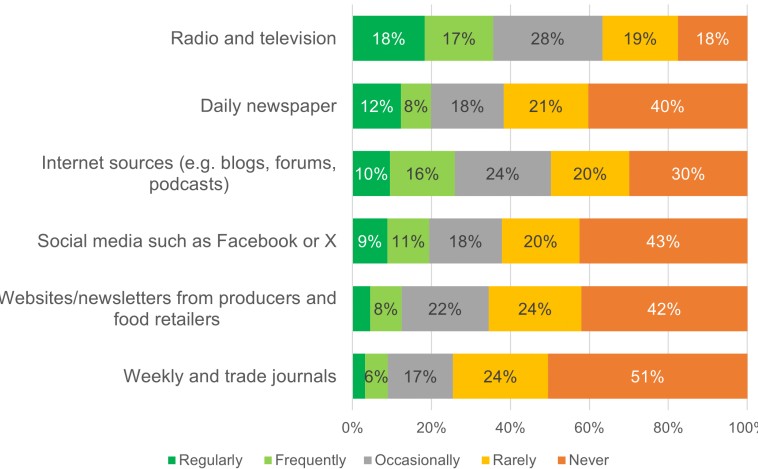

**Fig 6. Media use as a source of information for agriculture and food (n = 2,022), media format categories derived from [58].**

mean confidence, over- and underconfidence (following [53]), and the OLI (Table 7). Since the value of the individual over- and underconfidence is indirectly related to both, the correct knowledge rate and the confidence level, correlations can be assumed. A strong negative correlation between correct knowledge rate and over- to underconfidence (r=−0.779, *p<0.001*) indicates again that individuals with lower knowledge are more prone to overconfidence, while those with higher knowledge tend to be underconfident or better calibrated. There is a moderate positive correlation between mean confidence and over- and underconfidence (r=0.571, *p<0.001*), showing that as individuals' confidence increases, they are more likely to exhibit overconfidence. Additionally, mean confidence is positively correlated with opinion leadership (r=0.376, *p<0.001*), suggesting that opinion leaders tend to display higher confidence levels, regardless of their actual knowledge. Finally, a weak but significant positive correlation exists between over- and underconfidence and opinion leadership (r=0.233, *p<0.001*). However, there is no significant relationship between correct knowledge rate and opinion leadership, implying that being an opinion leader is not necessarily tied to actual knowledge levels. The findings underscore the importance of addressing overconfidence, particularly among opinion leaders, to improve the accuracy of information dissemination [60]. While the correlations identify meaningful statistical associations between accuracy, confidence, miscalibration, and opinion leadership, they cannot establish causal direction. For example, higher confidence may contribute to perceived leadership, but perceived leadership may also reinforce confidence. Thus, interpretations remain descriptive and should not be seen as evidence of causal mechanisms. The absence of a significant correlation between correct knowledge rate and the OLI is consistent with prior research showing that opinion leadership tends to be more strongly linked to communicative behaviour, self-confidence, and media use than to factual expertise. This reinforces the notion that opinion leaders in public debates are often influential

**Table 7. Correlation matrix between knowledge rate, confidence, and opinion leadership.**

| Correlation matrix | Correct knowledge rate | Mean confidence (all statements) | Over- and under-confidence [45] | Opinion Leadership (OLI) |
|---|---|---|---|---|
| Correct knowledge rate | 1 | | | |
| Mean confidence (all statements) | .068** | 1 | | |
| Over- and underconfidence | −.779*** | .571*** | 1 | |
| Opinion Leadership (OLI) | .004 | .376*** | .233*** | 1 |

Pearson: ***p<0.001; **p<0.01.

due to engagement and assertiveness rather than domain-specific knowledge. The positive correlation between miscalibration (over/underconfidence measure) and the OLI aligns with theories in social psychology suggesting that individuals who overestimate their abilities may participate more actively in discussions and assume leadership roles. This does not imply that overconfidence is beneficial; rather, it may increase communicative assertiveness and visibility.

### 3.3.2. Opinion leaders vs. strong opinion receivers.

The distribution of respondents based on the Opinion Leadership Index (OLI) revealed that only about one-fifth exhibited a positive index, classifying them as opinion leaders. Research on opinion leadership for the most part compares the first quartile ('opinion leaders') and the last quartile to explore differences in their traits and behaviours [6,61]. In this study, we compared the respondents in the highest and lowest quintiles ('strong opinion receivers') of the OLI to highlight the distinctive characteristics of opinion leaders by comparing the mean values of various measures (Table 8). The reason for comparing the two extreme quintiles instead of dividing the sample into four quantiles can be found in the overall distribution of individuals in the Opinion Leadership Index. (see Fig 5). The results indicate notable differences. Opinion leaders (Q1) scored significantly higher regarding the use of offline media formats as information sources (mean = 0.67, SD = 1.061) than opinion receivers (Q5, mean = −0.54, SD = 0.763), with a mean difference (MD) of 1.206. The confidence interval (CI) for this difference was 1.079–1.333, and the effect size (Cohen's d = 1.31) was strong. This suggests that opinion leaders engage more actively with traditional offline media. Similarly, opinion leaders also showed higher engagement with digital online media formats (Mean = 0.66, SD = 0.972) compared to opinion receivers (Mean = −0.55, SD = 0.799) (see also Munoz et al., 2022). Considering the six queried media formats individually, all six show statistically significant differences between Q1 and Q5. The average difference in usage frequency is lowest for social media platforms (MD = 1.300) and weekly or trade journals (MD = 1.433). In contrast, the difference is highest for internet sources such as forums, blogs, and podcasts (MD = 1.760). The latter can be explained by the more active role in information gathering that can be attributed to opinion leaders.

The comparison between the highest and lowest quintile (rather than quartile) is justified by the strongly left-skewed distribution of the OLI. Using quintiles ensures that the most pronounced opinion leaders (top 20%) and strongest opinion receivers (bottom 20%) are contrasted more sharply. Quartiles would include more "moderate" communicators and thus dilute the contrast between the extremes.

As already indicated in the correlation analysis, opinion leaders displayed a significantly greater confidence (Mean = 0.77, SD = 0.107) compared to receivers (Mean = 0.64, SD = 0.114) with a medium effect size (Cohen's d = 1.16).

**Table 8. Mean value comparison between 'opinion leaders' and 'strong opinion receivers'.**

| | Q | n | Mean | SD | F | Sig. | T | p | 95% CI (MD) | | | Cohen's d[a] | |
|---|---|---|---|---|---|---|---|---|---|---|---|---|---|
| | | | | | | | | | MD | LL | UL | Est. | r |
| Component 1: Classical offline media formats | Q1 | 404 | .67 | 1.061 | 67.747 | .000 | 18.635 | .000 | 1.206 | 1.079 | 1.333 | 1.31 | .55 |
| | Q5 | 404 | −.54 | .763 | | | | | | | | | |
| Component 2: Digital online media formats | Q1 | 404 | .66 | .972 | 22.730 | .000 | 19.399 | .000 | 1.121 | 1.092 | 1.337 | 1.37 | .56 |
| | Q5 | 404 | −.55 | .799 | | | | | | | | | |
| Mean confidence (all statements) | Q1 | 404 | .77 | .107 | 3.701 | .055 | 16.433 | .000 | −.128 | −.142 | −.113 | 1.16 | .50 |
| | Q5 | 404 | .64 | .114 | | | | | | | | | |
| Correct knowledge rate | Q1 | 404 | .64 | .156 | .718 | .397 | .780 | .218 | −.001 | −.031 | .016 | .06 | .02 |
| | Q5 | 404 | .63 | .164 | | | | | | | | | |
| Over- to under-confidence (J&A) | Q1 | 404 | .13 | .188 | .001 | .974 | −8.906 | .000 | −.119 | −.146 | −. 092 | .62 | .32 |
| | Q5 | 404 | .01 | .194 | | | | | | | | | |
| Self-assessment compared to the GP[b] | Q1 | 404 | 2.38 | .764 | 11.634 | .000 | 18.163 | .000 | −.899 | −.992 | −.803 | 1.28 | .54 |
| | Q5 | 404 | 1.49 | .636 | | | | | | | | | |

Note: Q1 = Quintile with highest OLI (opinion leaders); Q5 = Quintile with lowest OLI (strong opinion receivers);

[a]Cohen's d effect sizes: > .5 = medium; > .8 = strong; [b]categorical variable: p(two-sided) | Source: own survey.

This highlights again the opinion leaders' stronger self-assurance in their knowledge or opinions. In the same line, opinion leaders demonstrated higher overconfidence (Mean = 0.13, SD = 0.188) compared to strong opinion receivers (Mean = 0.01, SD = 0.194). The mean difference of −0.119 (95% CI: −0.146 to −0.092) and Cohen's d of 0.62 indicate a moderate effect, however, suggesting that opinion leaders tend to overestimate their abilities more frequently than their counterparts. As expected from the previous results, opinion leaders also scored higher in the self-assessment of their knowledge compared to the German population (Mean = 2.38, SD = 0.764) than receivers (Mean = 1.49, SD = 0.636), with a mean difference of −0.899 (95% CI: −0.992 to −0.803) and a large effect size (Cohen's d = 1.28). A pattern whereby individuals tend to overestimate informational deficiencies in others while perceiving themselves as comparatively well informed. Such self-assessments are known to directly influence information search behaviour [62]. Regarding the correct knowledge rate, the difference in correct answers between both groups was minimal (Q1: Mean = 0.64, SD = 0.156; Q5: Mean = 0.63, SD = 0.164), indicating no notable distinction in accuracy.

A comparison of polynomial functions (see 3.1.3 and S1 Table) between Opinion Leaders (Q1) and Strong Opinion Receivers (Q5) revealed that the Dunning-Kruger effect is significantly more pronounced among opinion leaders (F = 33.73; p = 0.0000). Opinion leaders, characterized by a generally higher level of self-confidence, tend to overestimate their abilities when their knowledge is limited.

Higher media engagement among opinion leaders is consistent with Two-Step and Multi-Step Flow theory, which conceptualises opinion leaders as individuals who consume more information and subsequently pass it on to others. Their higher usage of both, offline and online media therefore reflects their role as information intermediaries in public communication flows. Despite substantial differences in self-confidence and media use, opinion leaders do not differ from strong opinion receivers in terms of factual agricultural knowledge. This reinforces the earlier observation that opinion leadership is unrelated to expertise but strongly linked to communicative confidence and activity. Although opinion leaders exhibit a statistically stronger Dunning-Kruger-like pattern than strong opinion receivers, this should not be read as evidence for the full Dunning-Kruger effect. Instead, the result indicates that individuals with higher communicative confidence may also show stronger overconfidence when accuracy is lower, amplifying miscalibration patterns within this subgroup.

**3.3.3. Linear regression analysis: characterizing opinion leadership.** A linear regression analysis examines the relationship between the Opinion Leader Index (OLI) and various explanatory variables, as shown in Table 9. The unstandardized coefficient (B) indicates how much OLI changes with a one-unit change in the predictor, while the standard error (SE) reflects the precision of this estimate. The standardized coefficient (Beta) highlights the relative strength of each predictor in standard deviation units, enabling easy comparison across variables. Statistical significance is determined by p-values (Sig.), with smaller values, such as p < .001, indicating stronger significance. The 95% confidence interval (CI) provides the range within which the true effect size is likely to fall. Overall, the model explains 46.7% of the variance in OLI and is statistically significant (F(10, 2011) = 176.389, p < 0.01).

According to the regression model, several factors significantly influence the likelihood of being an opinion leader. The most impactful predictors are media usage and personal confidence. Both, classical offline media formats (B = .300, Beta = .388, Sig. < .001) and digital online media formats (B = .288, Beta = .373, Sig. < .001) showing highest Beta scores. Confidence also emerges as a critical factor, indicating that individuals with higher confidence levels are more likely to act as opinion leaders (B = 1.556, Beta = .236, Sig. < .001). Other important factors are personal contact with farmers and experience in agriculture, both of which can slightly increase the likelihood of opinion leadership and create a particularly positive image of agriculture (more so than exclusive contact via media; see Pfeiffer et al., 2021). Additionally, higher education levels (A-Levels, = German "Abitur" and higher) are positively associated with opinion leadership, albeit with a smaller effect (B = .066, Beta = .042, Sig. = .014) [63]. Gender has a small but significant effect, with males being slightly less likely to be opinion leaders (B = −0.70, Beta = −0.045, Sig. = .007). Age shows a weak, marginally significant negative association, suggesting that older individuals (over 50 years) might be less inclined to take on the role of opinion leader. These results on the influence of socio-demographic factors on knowledge accuracy and confidence concurs with other studies that have

**Table 9. Linear regression analysis to predict opinion leadership (n = 2,022).**

| Predictors for Opinion Leader (OLI) | B | SE | Beta | T | Sig. | 95% CI | |
|---|---|---|---|---|---|---|---|
| | | | | | | LL | UL |
| Mean knowledge rate | .025 | .083 | .005 | .300 | .764 | −.137 | .187 |
| Mean confidence (all nine statements)*** | 1.556 | .114 | .236 | 13.692 | <.001 | 1.333 | 1.779 |
| Comp. 1: Classical offline media formats (factor score)*** | .300 | .013 | .388 | 22.578 | <.001 | .274 | −.326 |
| Comp. 2: Digital online media formats (factor score)*** | .288 | .013 | .373 | 21.386 | <.001 | .262 | .314 |
| Gender (1 = male)** | −.070 | .026 | −.045 | −2.712 | .007 | −.121 | −.020 |
| Age (1 = +50 years)a | −0.52 | .028 | −.033 | −1.852 | .068 | −.107 | .004 |
| Education (1 = A-levels "Abitur" and higher)* | ,066 | .027 | .042 | 2.454 | .014 | .013 | .118 |
| Residential area size (1 = more than 20k inhabitants) | .036 | .026 | .023 | 1.388 | .165 | −.015 | .087 |
| Own agricultural experience (1 = yes)*** | .201 | .046 | .074 | 4.376 | <.001 | .111 | .292 |
| Personal contact with farmers (1 = yes)*** | .208 | .034 | .105 | 6.112 | <.001 | .141 | .274 |
| Constant*** | −1.730 | .094 | | −18.348 | <.001 | −1.915 | −1.545 |

Note: ***$p < .001$. **$p < 0.01$, *$p < 0.05$, a$p < 0.1$; $R^2 = .467$, corr. $R^2 = .465$, $F(10, 2011) = 176.389$, $p < .001$ | Source: own survey

carried out group comparisons [45,64]. The residential area size does not significantly influence opinion leadership. Again, opinion leaders cannot be identified via their knowledge level, and this result supports the disconnect between actual knowledge and opinion leadership, which has already been noted in the results presented above (see also [65]).

However, media-use factors and confidence clearly emerged as the strongest predictors of the OLI, whereas factual knowledge was not significant. This underscores that opinion leadership in this context is driven less by expertise and more by communicative behaviour, exposure to information sources, and subjective certainty. Sociodemographic predictors showed only modest effects: age and gender had small negative associations with opinion leadership, and higher education had a weak positive effect. These results align with previous communication studies, which often find that demographic variables play a secondary role compared to psychological and behavioural factors. An interaction term between knowledge accuracy and mean confidence was tested but did not contribute significantly to the model ($p > 0.05$). This indicates that the relationship between confidence and opinion leadership does not depend on actual knowledge levels. While the model explains a considerable share of variance (≈47%), the estimates reflect associations rather than causal effects. Opinion leadership is a complex construct that likely emerges from psychological, social, and contextual influences not fully captured by the present model.

## 4. Conclusions

This study investigated the critical role of opinion leaders in shaping public understanding of agricultural issues, focusing on their knowledge accuracy, self-confidence, and media engagement. Using data from a representative survey of the German population, this research examined the interplay between cognitive biases, such as the Dunning-Kruger Effect, and the behaviours of opinion leaders within the framework of the Two-Step Flow Communication Model and Social Cognitive Theory. The findings figure out the level of actual knowledge accuracy and self-assessed confidence and offer nuanced insights into the characteristics, strengths, and limitations of opinion leaders and provide directions for practical and policy interventions. Although the study identifies statistically meaningful associations between knowledge, confidence, media use, and opinion-leadership tendencies, these relationships are descriptive and should not be interpreted as causal mechanisms. According to the survey data, opinion leaders within the non-agricultural German population stand out for their high levels of confidence compared to the general population, but this confidence is not consistently aligned with superior knowledge accuracy. Social Cognitive Theory highlights the role of overconfidence as a cognitive bias,

particularly in complex domains like agriculture and food production. While some patterns resemble elements of the Dunning-Kruger discussion, the results should not be taken as evidence for the full canonical Dunning-Kruger curve. Instead, they indicate structured miscalibration patterns that vary with item difficulty and confidence levels. The findings should be interpreted as descriptive associations rather than evidence of causal effects, as the study does not measure actual information transmission, audience reception, or behavioural consequences.

Remarkably, opinion leaders often engage more actively with both, traditional and digital media platforms, positioning them as influential nodes in the information dissemination network. However, their media consumption does not necessarily guarantee the accuracy of the information they relay, as their sources may sometimes be incomplete, distorted, or unsuitable. This underscores that the influence of opinion leaders depends not only on their communicative activity but also on the quality, credibility, and completeness of the media sources they rely on.

One of the most striking findings is the lack of a significant relationship between knowledge accuracy and opinion leadership. This suggests that while opinion leaders are influential in shaping public perceptions, their influence is not inherently tied to their level of factual knowledge. Instead, their self-assurance and ability to communicate confidently appear to play a more substantial role. This result aligns with longstanding findings in communication science that opinion leadership is more strongly associated with communicative assertiveness, social embeddedness, and media engagement than with domain expertise. This contradiction underscores the importance of addressing cognitive biases and enhancing the quality of the information these leaders disseminate. The study also demonstrates behavioural and perceptual differences between opinion leaders and their audiences as 'opinion receivers'. Opinion leaders (on the consumer level) exhibit greater engagement with media, higher levels of confidence, and a stronger tendency towards overconfidence. While their self-assessment abilities are generally paired with higher self-confidence, their actual knowledge accuracy is only marginally higher than that of opinion receivers. These results should not be read as identifying opinion leaders as systematically misinformed but rather as highlighting the divergence between perceived competence and measured knowledge—a common phenomenon in public discourse on technical topics. The nuanced understanding of opinion leadership traits reveals both, the opportunities and challenges in leveraging their influence for societal benefit. While this pattern is not entirely novel and aligns with findings from broader communication and consumer research, its replication within the context of agricultural knowledge and a representative general population sample remains rare. The contribution of this study therefore lies not in identifying a new mechanism, but in empirically demonstrating the persistence of confidence-driven opinion leadership in a domain characterized by high societal relevance and increasing misinformation risks.

From an applied perspective, these findings offer several pathways for improving public communication and education. Training initiatives should focus on helping individuals, especially opinion leaders, recognise and adjust overconfidence. Suitable initiatives may include confidence calibration exercises, structured feedback formats, or interactive knowledge-assessment modules that help individuals better align perceived with actual knowledge. Such programmes could target secondary schools, farmers' associations, and community-organised groups to develop critical thinking and self-assessment skills. Workshops and informational exchange formats on agriculture and food designed specifically for 'first level' opinion leaders—such as bloggers, influencers, and other digital communicators—should aim to provide validated, accessible knowledge and tools for critically evaluating their sources. Integrating media literacy components, such as how to detect misinformation, evaluate source credibility, and verify claims, may substantially improve the reliability of the information passed along by opinion leaders. Empowering these individuals to distinguish reliable information from misinformation can significantly enhance the quality of public discourse. Leveraging both, traditional and digital media platforms, such as Facebook, TikTok, and LinkedIn, can help reach diverse audiences.

These platforms differ substantially in algorithmic amplification, audience fragmentation, and message formats; thus, communication strategies should be tailored to the affordances and risks of each platform. However, ensuring the scientific accuracy and engagement of the content is vital. Communication strategies should also focus on influencing those who generate content for these platforms to prevent the initial spread of misinformation. Designing communication efforts

that address the cognitive profiles of various audience segments can help bridge knowledge gaps. Customised communications and communication strategies that target cognitive biases and self-efficacy of individual populations can improve the resonance and impact of messages.

The results further underscore the importance of policy-level interventions to harness the potential of opinion leaders while mitigating their limitations. Identifying and collaborating with opinion leaders who are active in blogs and online forums or have personal ties to agriculture can amplify initiatives promoting sustainable practices. This approach aligns with recommendations in agricultural communication research, which emphasise working with trusted multipliers who already act as informal knowledge brokers in their communities. Providing these leaders with access to credible, scientifically validated resources increases the likelihood that they relay high-quality information. Implementing systems that provide constructive feedback to opinion leaders and their audiences can help recalibrate overconfidence and promote more accurate knowledge dissemination. These may include moderated discussion forums, expert Q&A formats, or automated fact-checking prompts integrated into social media platforms.

Thematic web portals run by independent and public organisations that check facts and dispel half-truths and myths are good examples that can be specifically targeted at opinion leaders. Educating the public and opinion leaders about the effects of cognitive biases, such as the Dunning-Kruger Effect, can improve their understanding of the limits of their knowledge and decision-making processes. Establishing dialogue-oriented platforms where experts, opinion leaders, and the public can interact, encourages collaborative learning and the co-creation of knowledge-sharing programmes. Involving local communities, such as municipal groups or community-led initiatives, ensures that these programmes are culturally and contextually relevant.

While this study provides valuable insights, it also highlights areas for future exploration. Understanding cross-cultural differences in opinion leadership dynamics could enhance the global applicability of these findings. The generalisability of the present findings is limited by the focus on one national context and by the structured nature of online panel surveys, which, despite quota sampling, may underrepresent less digitally active population groups. Investigating whether traditional and digital media engagement leads to distinct knowledge outcomes is another crucial avenue. Moreover, exploring the emotional and personal relevance of topics, such as nutrition or health, could reveal differences in how opinion leaders engage with and disseminate information about agricultural processes and food production. Addressing the challenge of reaching opinion leaders and correcting pre-existing misconceptions remains a critical research question. How can we influence opinion leaders to reconsider their beliefs and align them with evidence-based knowledge? Additionally, it is essential to examine the extent to which opinion leaders' beliefs align with the information they pass on and how to ensure the accuracy of their messaging. Opinion leaders, though not inherently more knowledgeable, possess significant influence as social connectors. Their role in shaping public understanding of critical issues like agriculture can be leveraged to promote evidence-based practices, provided they receive appropriate guidance and resources. Ultimately, improving public discourse requires not only equipping opinion leaders with accurate information but also fostering environments in which they are encouraged to reflect critically on their role, question their assumptions, and engage constructively with expert knowledge. By addressing cognitive biases, calibrating confidence levels, and equipping opinion leaders with accurate, validated information, we can enhance their positive impact on public discourse. These efforts are vital not only for improving agricultural communication but also for advancing public understanding in related domains such as food processing, healthy nutrition, and sustainability.

## Supporting information

**S1 Table. DKE mean values and curve progressions – Interplay of correct knowledge rate and confidence levels in different sample segments.**
(DOCX)

**S2 Dataset. Anonymized raw survey data (Excel file) including all variables and labels used in the analyses.**
(XLSX)

## Acknowledgments

We would like to thank the project members of the Future Crop Farming (A) research and demonstration project and several agricultural experts from the Bavarian State Research Centre for Agriculture for their support in planning and conducting this study.

## Author contributions

**Conceptualization:** Andreas Gabriel.

**Data curation:** Andreas Gabriel.

**Formal analysis:** Andreas Gabriel.

**Investigation:** Andreas Gabriel.

**Methodology:** Andreas Gabriel.

**Project administration:** Andreas Gabriel.

**Resources:** Andreas Gabriel.

**Software:** Andreas Gabriel.

**Validation:** Andreas Gabriel.

**Visualization:** Andreas Gabriel.

**Writing – original draft:** Andreas Gabriel.

**Writing – review & editing:** Vera Bitsch.

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
