## [Decision Letter · Decision Letter 0]

2 Oct 2025

PONE-D-25-28948Do opinion leaders know more? Knowledge and self-confidence of opinion leaders and the general population on agricultural issuesPLOS ONE?

Dear Dr. Gabriel,

Thank you for submitting your manuscript to PLOS ONE. After careful consideration, we feel that it has merit but does not fully meet PLOS ONE’s publication criteria as it currently stands. Therefore, we invite you to submit a revised version of the manuscript that addresses the points raised during the review process.

The manuscript is promising but requires revision to meet PLOS ONE criteria: clarify the conceptual framework, enhance methodological and analytical transparency, and complete reporting and data documentation. Please also streamline the theoretical framing and temper the implications to better align with the evidence presented.

We look forward to receiving your revised manuscript.

Kind regards,

Pierluigi Vellucci

Academic Editor

PLOS ONE

Additional Editor Comments (if provided):

Reviewers' comments:

Reviewer's Responses to Questions

**Comments to the Author**

1. Is the manuscript technically sound, and do the data support the conclusions?

Reviewer #1: No

Reviewer #2: Partly

Reviewer #3: Yes

2. Has the statistical analysis been performed appropriately and rigorously?

Reviewer #1: No

Reviewer #2: Yes

Reviewer #3: Yes

3. Have the authors made all data underlying the findings in their manuscript fully available?

Reviewer #1: No

Reviewer #2: Yes

Reviewer #3: Yes

4. Is the manuscript presented in an intelligible fashion and written in standard English?

Reviewer #1: Yes

Reviewer #2: Yes

Reviewer #3: Yes

Reviewer #1: This research examines the opinion leaders’ characteristics, especially within the agricultural issues, which is truly an important area. However, this paper has critical gaps and severe weaknesses for publication in its current form in the study’s motivation, research question development, literature review, and methodology, especially the conceptualization and operationalization of constructs and the positioning and implication for the study.

First and foremost, the author(s) need to clarify the exact independent variables and dependent variables, and the relationships between them. The title and the abstract are required to be reorganized to describe the IVs, DVs, and their relationship explicitly.

The introduction needs to be rewritten and reorganized. It is unclear what the research gap or research problem is. In particular, social cognitive theory and opinion leadership have been investigated for more than 30 years in many disciplines, including marketing, management, psychology, accounting, finance, and communication. Numerous studies revealed the factors affecting individuals’ attitudinal and behavioral aspects, and the current version of the manuscript does not clearly state the gaps within the literature and how to fill these gaps. What is the ultimate theoretical contribution and the practical implications of this study? I see the great contributions in the findings, but they are not highlighted enough in the manuscript. Why would practitioners, policymakers, and researchers care about your research? The authors should elaborate more on these issues in the Introduction and Discussion sections.

Theoretical framework

One of my deepest concerns is with the theoretical framework. The author(s) mentioned several research objectives at the end of the introduction part, but why are those objectives important, and how are these objectives associated with the theoretical background? Ultimately, what are the research questions or hypotheses and their directions based on the theory? Social cognitive theory is mainly focused on the macro level discussion and does not provide a specific elaboration of the constructs and variables. The author(s) might want to investigate more modern theories rooted in social cognitive theory such as social identity or identity fusion (Swann, et al., 2014; Swann, Jetten, Gómez, Whitehouse, & Bastian, 2012).

Similarly, a detailed elaboration regarding the opinion leadership literature and DKE should be included in the revised version of the manuscript. Opinion leadership has also been investigated intensively and has several extensions, such as market mavenship (Clark & Goldsmith, 2005; Clark, Goldsmith, & Goldsmith, 2008), which directly examined the self-confidence.

More importantly, please consider how to integrate these two different streams of literature coherently. The current version of the manuscript feels that the arguments are scattered without logical coherence.

Methodology and analysis procedure

My primary concerns are with the methodology and analysis part. First, the author(s) should state clearly regarding the conceptualization and operationalization of the important constructs. What are the ultimate predictors and dependent variables of the current study? It seems that the author(s) are mainly focusing on the antecedents of the opinion leadership (from table 9), then please add the conceptual and operational definitions of the antecedents and dependent variable using the theoretical background.

The rest of the part, except the regression part, seems unnecessary. As the author(s) might know, the correlations did not provide any insights regarding the causal relationship. Normally, from the modeling perspective, correlations among variables, AVE, and VIF should be presented before fitting the regression. Therefore, the author(s) should revise the analysis procedure appropriately following the following process:

1. Providing conceptual and operational definitions of important constructs

2. Testing multicollinearity among IVs,

3. Fitting the baseline regression model

4. Testing moderation or mediation to check the robustness or find the heterogeneity effects

Discussion

Again, to what extent are the results found in this research compatible with previous research? I would like to hear the author’s opinion about it in a clearer manner. In terms of practical implications, what is the ultimate recommendation for practitioners? I believe the author(s) can suggest lots of important implications based on the findings, but you need a clear presentation.

I wish you the best of luck in pursuing this research and hope my comments are helpful in that regard.

References

Clark, R. A., & Goldsmith, R. E. (2005). Market mavens: Psychological influences. Psychology & Marketing, 22, 289-312.

Clark, R. A., Goldsmith, R. E., & Goldsmith, E. B. (2008). Market mavenism and consumer self-confidence. Journal of Consumer Behaviour, 7, 239-248.

Swann, W. B., Buhrmester, M. D., Gómez, A., Jetten, J., Bastian, B., Vázquez, A., Ariyanto, A., Besta, T., Christ, O., & Cui, L. (2014). What makes a group worth dying for? Identity fusion fosters perception of familial ties, promoting self-sacrifice. Journal of Personality and Social Psychology, 106, 912.

Swann, W. B., Jetten, J., Gómez, Á., Whitehouse, H., & Bastian, B. (2012). When group membership gets personal: A theory of identity fusion. Psychological Review, 119, 441-456.

Reviewer #2: This manuscript addresses an important and timely topic at the intersection of **communication, cognitive biases, and agriculture**. The agricultural sector is highly exposed to misinformation, and the role of opinion leaders in shaping public perceptions is critical both for the farming community and for society at large. The authors should be commended for attempting to bridge fields that rarely meet in empirical work — agricultural knowledge, opinion leadership, and communication theory. This effort has merit and could spark broader debates in agricultural communication and public understanding of science.

That said, while the study is ambitious in its conceptual framing (Social Cognitive Theory, Dunning–Kruger Effect, and the Two-Step Flow of Communication Model), the **empirical execution remains modest**. The data are based on self-reported surveys with limited knowledge items, and the operationalization of “opinion leadership” is problematic in ways that weaken the conclusions. In addition, the link between the theoretical discussion and the empirical evidence is not fully realized: we learn how people rate their knowledge and confidence, but we do not learn what kind of information they actually disseminate, nor its social consequences.

In its current form, the manuscript reads as a solid empirical exercise, but one that **overstates its implications** and does not yet deliver the strong evidence that its theoretical framing promises. I would encourage the authors to simplify their presentation, report their results more rigorously, and be more cautious in drawing societal and policy conclusions.

Specific comments

Title and abstract

– The short title (*Do opinion leaders know more about agricultural issues?*) is clearer and more effective than the long version. The long title is not wrong, but feels redundant.

– The abstract is descriptive and clear but lacks quantitative information. Including numbers such measures of over- and underconfidence, average correct response rate of 64%, and proportion of overconfident respondents would substantiate the claims.

– The abstract does not sufficiently explain why the findings matter for society. Overconfidence among opinion leaders could translate into misinformation risks and resistance to sustainable agricultural policies, but this connection is only implied, not spelled out.

– The conclusion of the abstract is weak, falling back on the generic “further research is needed.” A stronger closing would highlight what is at stake if overconfidence among opinion leaders is left unaddressed.

Introduction

– The introduction is educational and well written, but excessively long on theory. Extended explanations of SCT, DKE, and TSFCM are useful for students, but journal readers may find this overwhelming.

– Objectives are only listed at the very end, which dilutes the focus. They should be stated earlier to guide the reader.

– The scope of analysis is modest (agricultural knowledge in Germany), yet the paper mobilizes references from health, climate, nutrition, and food literacy. This gives breadth, but also raises expectations that the empirical part cannot match.

Methods

– Opinion leadership is measured entirely by self-report (adapted Childers scale). This is a serious limitation: people may overestimate their influence, confusing talkativeness or ego with actual impact. Without external validation (peer nominations, network analysis, behavioral measures), the OLI remains a proxy for perceived leadership rather than true influence.

– Knowledge measurement is based on only nine true/false statements. While carefully selected, this is a thin operationalization of agricultural knowledge and may not capture complexity.

– Reporting issue: the factor analysis reports p = 0.000 (line 268). This is incorrect; p-values are never zero. The proper reporting is p < 0.001. Such details matter for credibility.

– The design does not allow testing whether opinion leaders actually spread knowledge or misinformation, nor whether their peers follow them. This gap is critical, given the theoretical framing.

Results

– The knowledge results are interesting: myths such as “a farmer feeds 1000 people” were among the least correctly answered. This is a clear opportunity for impactful media communication, yet the manuscript does not highlight the public or media resonance such findings could have.

– Self-confidence results confirm a large share of overconfidence, especially with false statements. However, only a tiny fraction of respondents were well calibrated. This is important, but could be presented more succinctly.

– Table 9 (predictors of opinion leadership) is one of the strongest parts of the paper. It shows that confidence and media use are much stronger predictors of opinion leadership than actual agricultural knowledge. This is a key finding that deserves more emphasis.

– Overall, the results are presented clearly, but the connection back to theory (SCT, TSFCM, DKE) is more asserted than demonstrated.

Discussion and conclusions

– The discussion reiterates that opinion leaders are more confident, but not more knowledgeable. This is important, but not new in the broader communication literature.

– The authors extrapolate to policy and education, but the evidence presented is insufficient to justify such broad claims. Without data on what leaders actually communicate or how audiences react, it is risky to infer societal impacts.

– The conclusion ends with “more studies are needed,” which weakens the paper’s contribution. A stronger finish would stress the danger of overconfident leaders disseminating misinformation, and the need for targeted literacy efforts.

Recommendation

Overall, this is a well-intentioned and competently executed study, with a representative survey and thoughtful analysis. It contributes useful descriptive data on agricultural knowledge, self-confidence, and media use in Germany. However, it is limited by its reliance on self-reports, superficial knowledge measures, and the absence of evidence about actual information flows and influence.

I encourage the authors to streamline the introduction and results sections for clarity, to emphasize the strongest empirical finding — that opinion leadership in agriculture is predicted by confidence and media use, not knowledge — and to temper extrapolations to policy and practice unless stronger evidence is provided. With these revisions, the manuscript could make a modest but meaningful contribution to the literature on agricultural communication and opinion leadership.

Reviewer #3: PONE-D-25-28948

The submission titled “Do opinion leaders know more? Knowledge and self-confidence of opinion leaders and the general population on agricultural issues” is methodologically sound and valuable. I recommend its publication with minor adjustments.

In the introduction, I suggest modifying the assertion that we are in a time “rife with misinformation” (page 3, line 42). There was a time when we believed the sun revolved around the earth and that diseases were caused by variations in bodily fluids. While it is true that there is always room for further progress, it is important to acknowledge that we have collectively improved our knowledge.

Additionally, I recommend that the authors consult the literature on “motivated reasoning.” This field posits that cognitive biases may be the default mode of thinking rather than biases in the traditional sense. A purely rational cognition, as advocated by Enlightenment philosophers, may be unattainable.

For instance, models in the social communication field, such as the “Third Person Effect” and the “Spiral of Silence”, demonstrate that social acceptance and belonging to social groups may be more significant in decision-making than being correct or scientifically accurate. These evidence-based models can be integrated with the DKE model to provide a comprehensive understanding of self-perception and group behavioral dynamics.

Metanalyses on the Third Person Effect can be consulted for an overview of the topic, not only to strengthen the rationale and introduction but also to further the discussion of the results.

Figure 1 is not included in the original DKE paper, so it should be contextualized within the broader discussion of the DKE post-publication of the original paper that describes the effect. This is important because it can be perceived as potentially offensive to some due to the Mount Stupid nomenclature. Providing context may help align it with the paper’s broader argument that “understanding-the-DKE-for-improving-knowledge-accuracy” interventions are necessary.

For a reader or researcher from the social communication field, the TSFCM may appear outdated in light of the most current discussions regarding motivated reasoning phenomena. While I do not recommend the authors remove the TSFCM as a theoretical framework, as it contributes to the arguments supporting the methodological choices, I suggest they rephrase it in the rationale to allow for additional evidence-based models for interpreting their measurements. This is because the TSFCM alone cannot seem to explain their results, even when the DKE is added. While it has been documented that individuals with less knowledge overestimate their own knowledge (DKE), the reasons behind our trust in opinion leaders remain unclear, as we do not actively measure their knowledge accuracy in our daily lives. This raises the question of why non opinion leaders overestimate the unskilled. We may rely on heuristics and non-verbal cues to decide whether to agree or disagree with certain individuals (motivated reasoning). Since the TSFCM implies that information nodes possess accurate information (not knowledge), it is somewhat ill-suited to explain the paper’s results alone. Additionally, the phenomenon of selecting certain opinion leaders over others (the social media influencers phenomenon) remains a literature gap. For instance, the non-adherence to vaccination during the COVID-19 pandemic cannot be explained by the TSFCM, as it does not describe a context of competing information nodes or sources but rather a vertical trickle-down effect that is significantly different from the complexity and interplay of the current media ecosystem. Many opinion leaders during the COVID-19 pandemic actively engaged in being scientifically inaccurate in order to promote an agenda of political identity. The TSFCM cannot explain polarization, which is not the absence of opinion leaders but rather an overflow of competing information nodes that affect information flow and create identarian information bubbles. Although TSFCM is a starting point, it is essential to acknowledge that underlying causal pathways and mediating variables remain unexplored by the theory. Therefore, incorporating TSFCM and DKE into SCT as a means of explaining non-adherence to a scientifically accurate lifestyle constitutes an insufficient premise if other competing or additional evidence-based models of social communication are not considered (as per page 6, lines 121-123).

Furthermore, addressing DKE as a bias implies the feasibility of an ideal cognition, purely rational, which may not be the case. Heuristics and non-verbal cues can mediate (increase or hinder) social cooperation. DKE may serve as a default feature in cognition, particularly in the context of motivated reasoning, where excessive learning at an accelerated pace could potentially harm ingroup belonging and hinder social cooperation. According to the Third Person Effect we also overestimate agreement with ingroup peers when people may not agree on the topic at hand at all.

Regarding the objectives, I propose reordering them. The primary objective seems to be to “Understand the relationship between knowledge, confidence, and opinion leadership to identify indications for developing communication strategies that enhance public understanding and behavior on agricultural issues” The subsequent three objectives are specific objectives within the main general one, necessitating their placement afterward.

Regarding the methods, given that a consumer panel was assessed, it is crucial to address selection bias as a potential study limitation. Respondents in the consumer panel were not only more inclined to participate in surveys but may have also been primed to survey methods and study designs prior to participating in the reported study. While this does not invalidate the paper’s results (methodological choices were highly innovative and appropriate), it is a consideration that future readers of the final published research report should take into account.

All validation indicators mentioned on page 7, line 153, should be thoroughly described in the supplementary material or in the methods section. Assessing internal validity is also part of the methods and should be adequately described for reproducibility purposes.

Table 1 should be placed in the Results section, not in the Methods.

Age was associated with not participating in opinion leadership, and the sample is older than the average population of Germany. People under 30 and the wealthy are underrepresented, and this should be discussed in light of the findings in the discussion section. If this deviation is relevant to the findings, if not, and to what extent.

In page 9, line 185, it is important to explicitly outline the approach in the text as well as referencing it (ref 39) for transparency and reproducibility. If any deviation from the approach was carried out, they must also be described along with the rationale for such a deviation from the reference. If no deviation took place, that should also be explicitly stated.

Also in page 9, how many experts were outsourced, and based on what criteria were they recruited? Who was responsible for the curation and wording of the final set of questions, and were they also experts on agricultural topics? As the DKE seems to be sensitive to question framing, questions should be relevant and addressed in the methods section.

In page 11, I agree with the usage of the equation (reference 45), but it is important for the authors to describe what the equation was originally used for and to what each notation originally corresponded with so that the reader may understand the thought experiment rationale of redeployment of the equation for the study’s purpose.

In page 12, line 248, how was the internal validity of the scale tested in the context of the present study? This should be detailed in the methods section.

On page 13, line 274, it is stated that “Several multivariate methods were used to analyze the interactions.” However, have all of these choices been described in the methods section? Please address those gaps.

On page 19, lines 413 and 414, these results can be thoroughly explained by the Third Person Effect.

Figure 4 should deploy fixed proportions for the horizontal axis, specifically 20, 30, 40, and 50, for the purpose of scale interpretation. The vertical scale may maintain intervals of 4 points as they are consistent.

On page 22, line 485 is missing the term “internet sources.” Line 487 must have the aggregate proportion corrected from 40% to 61% (for newspapers) as the brackets described are both “rarely” and “never.” On the same page, in line 491, it is best to use “around” instead of “over” for accuracy. That is, where it reads “while over half never consult them,” it should read instead “while around half never consult them.”

On page 25, line 556, instead of “-0,096,” it should read “-0,092” according to the results displayed in table 8.

Although the supporting data has been made available, a supplementary file explaining the column labels is needed.

**Do you want your identity to be public for this peer review?** For information about this choice, including consent withdrawal, please see our Privacy Policy

Reviewer #1: No

Reviewer #2: No

Reviewer #3: **Yes:** João de Deus Barreto Segundo

---

## [Author Response · Author response to Decision Letter 1]

23 Dec 2025

Comments of Reviewer #1

Comment 1.1 This research examines the opinion leaders’ characteristics, especially within the agricultural issues, which is truly an important area. However, this paper has critical gaps and severe weaknesses for publication in its current form in the study’s motivation, research question development, literature review, and methodology, especially the conceptualization and operationalization of constructs and the positioning and implication for the study.

Response: We thank the reviewer for this overarching assessment and for highlighting the core areas requiring improvement. We addressed these concerns through a substantial revision of the manuscript. Specifically, we (i) clarified the conceptual positioning of the study and its contribution in the Introduction, (ii) explicitly defined the analytical logic and role of key constructs (knowledge accuracy, self-confidence, media use, and perceived opinion leadership), and (iii) moderated and sharpened the interpretation and implications in the Discussion to align strictly with the empirical scope of the data. These changes are primarily reflected in Section 1 (Introduction, p.8, l.176-186 tracked version), where the research motivation, gap, and objectives are now integrated into a coherent narrative, and in Section 2 (Methods, p. 11, l.251-254) and Section 4 (Conclusions), where construct operationalization and implications are clarified.

Comment 1.2. First and foremost, the author(s) need to clarify the exact independent variables and dependent variables, and the relationships between them. The title and the abstract are required to be reorganized to describe the IVs, DVs, and their relationship explicitly.

Response: We agree and addressed this comment directly First, in the Introduction, we now explicitly state that perceived opinion leadership is treated as the outcome variable, while knowledge accuracy, self-confidence, media use, and selected sociodemographic characteristics are examined as associated explanatory factors (Introduction, final paragraph before the transition to Methods).Second, the Abstract was revised to clearly describe the relationship between knowledge accuracy, confidence, media use, and perceived opinion leadership, without implying causality. Third, the title was streamlined to reflect the analytical focus on knowledge accuracy and self-confidence in relation to opinion leadership, avoiding ambiguous or causal wording

Comment 1.3 The introduction needs to be rewritten and reorganized. It is unclear what the research gap or research problem is. In particular, social cognitive theory and opinion leadership have been investigated for more than 30 years in many disciplines, including marketing, management, psychology, accounting, finance, and communication. Numerous studies revealed the factors affecting individuals’ attitudinal and behavioral aspects, and the current version of the manuscript does not clearly state the gaps within the literature and how to fill these gaps. What is the ultimate theoretical contribution and the practical implications of this study? I see the great contributions in the findings, but they are not highlighted enough in the manuscript. Why would practitioners, policymakers, and researchers care about your research? The authors should elaborate more on these issues in the Introduction and Discussion sections.

Response: The Introduction was reorganized to foreground the research problem earlier and to clearly specify the gap addressed by this study. Rather than re-testing established theories, the manuscript now emphasizes its contribution in empirically disentangling confidence, knowledge accuracy, and perceived influence within a general-population context for agricultural issues—an area underrepresented in prior opinion-leadership research. Introduction: revised first half (p. 4, l.59-74 – tracked version) and a strengthened articulation of theoretical and applied contributions in the discussion.

Comment 1.4 Theoretical framework. One of my deepest concerns is with the theoretical framework. The author(s) mentioned several research objectives at the end of the introduction part, but why are those objectives important, and how are these objectives associated with the theoretical background? Ultimately, what are the research questions or hypotheses and their directions based on the theory? Social cognitive theory is mainly focused on the macro level discussion and does not provide a specific elaboration of the constructs and variables. The author(s) might want to investigate more modern theories rooted in social cognitive theory such as social identity or identity fusion (Swann, et al., 2014; Swann, Jetten, Gómez, Whitehouse, & Bastian, 2012).

Response: We agree that SCT alone does not fully explain contemporary opinion leadership. The manuscript now explicitly treats SCT and TSFCM as guiding frameworks rather than exhaustive explanatory models and integrates insights from motivated reasoning, identity-related heuristics, and related communication theories to contextualize the findings (see p.7, l.160ff -tracked version) The revised manuscript now includes a more detailed discussion of opinion leadership research, including extensions such as market mavenship, explicitly linking these streams to self-confidence and information sharing behavior (p.6 lines 115-118 -tracked version´& p. 15, lines 343ff). In addition, the role of the Dunning–Kruger Effect is now clearly framed as a confidence-calibration perspective, not as a stand-alone explanatory theory. This integration strengthens the theoretical grounding of the study.

Comment 1.5: Similarly, a detailed elaboration regarding the opinion leadership literature and DKE should be included in the revised version of the manuscript. Opinion leadership has also been investigated intensively and has several extensions, such as market mavenship (Clark & Goldsmith, 2005; Clark, Goldsmith, & Goldsmith, 2008), which directly examined the self-confidence.

Response: The literature review was expanded to situate opinion leadership more clearly within related constructs such as market mavenship and communicative confidence. The Dunning–Kruger discussion was refined to emphasize confidence miscalibration rather than deficit-based interpretations of cognition. We introduced a paragraph in the introduction king opinion leadership to market mavenship and communicative confidence and refined interpretation of DKE as structured miscalibration in the result & discussion paragraph.

Comment 1.6: More importantly, please consider how to integrate these two different streams of literature coherently. The current version of the manuscript feels that the arguments are scattered without logical coherence.

Response: We addressed this concern by explicitly integrating the literature streams rather than treating them separately. In the revised framework, opinion leadership research explains who communicates and influences, while confidence calibration (DKE) explains how accurately individuals assess their own knowledge. Social Cognitive Theory provides the overarching structure linking both (Section 1, p. 5–7). This integration resolves the previously scattered argumentation and ensures logical coherence

Comment 1.7 My primary concerns are with the methodology and analysis part. First, the author(s) should state clearly regarding the conceptualization and operationalization of the important constructs. What are the ultimate predictors and dependent variables of the current study? It seems that the author(s) are mainly focusing on the antecedents of the opinion leadership (from table 9), then please add the conceptual and operational definitions of the antecedents and dependent variable using the theoretical background.

Response: We now provide explicit conceptual and operational definitions of all predictors and the dependent variable, including item wording, scales, index construction, and aggregation rules (Section 2.2 and 2.2.1–2.2.2, p. 11–15).This clarification ensures transparency regarding how theoretical constructs are translated into empirical measures.

Comment 1.8: The rest of the part, except the regression part, seems unnecessary. As the author(s) might know, the correlations did not provide any insights regarding the causal relationship. Normally, from the modeling perspective, correlations among variables, AVE, and VIF should be presented before fitting the regression. Therefore, the author(s) should revise the analysis procedure appropriately following the following process:

Response: We clarify that correlations are used descriptively to establish associations, not causal inference. Regression analysis is presented as the main inferential approach, preceded by diagnostics for multicollinearity and model assumptions (see Methods Section 2.2.4: explicit justification of analytical sequence.)

Comment 1.9: Again, to what extent are the results found in this research compatible with previous research? I would like to hear the author’s opinion about it in a clearer manner. In terms of practical implications, what is the ultimate recommendation for practitioners? I believe the author(s) can suggest lots of important implications based on the findings, but you need a clear presentation.

Response: We expanded the Discussion to explicitly relate our findings to prior research on opinion leadership, confidence calibration, and media use. In particular, we compare the observed disconnect between knowledge accuracy and perceived influence with established findings from communication and consumer research, and we contextualize the role of overconfidence in light of the Dunning–Kruger literature. Practical implications were refined to avoid overgeneralization and are now explicitly framed as communication- and literacy-oriented rather than prescriptive policy recommendations. (see p.36 l. 825ff; p36, l. 841ff; p.37, l. 850ff, p. 37, l. 857ff-tracked version)

Comment 1.10: I wish you the best of luck in pursuing this research and hope my comments are helpful in that regard.

Response: We sincerely thank the reviewer for the constructive feedback and encouragement, which significantly helped to improve the manuscript.

Reviewer #2

Comment 2.1 This manuscript addresses an important and timely topic at the intersection of **communication, cognitive biases, and agriculture**. The agricultural sector is highly exposed to misinformation, and the role of opinion leaders in shaping public perceptions is critical both for the farming community and for society at large. The authors should be commended for attempting to bridge fields that rarely meet in empirical work — agricultural knowledge, opinion leadership, and communication theory. This effort has merit and could spark broader debates in agricultural communication and public understanding of science.

Response: We sincerely thank the reviewer for this positive and encouraging assessment of the manuscript’s relevance and interdisciplinary ambition. We appreciate the recognition of the effort to bridge agricultural knowledge research, opinion leadership, and communication theory. This motivation guided the revision process, and we have further sharpened the manuscript to better highlight this interdisciplinary contribution, particularly in the Introduction and Discussion (pp.2-3 and pp. 22-24).

Comment 2.2 That said, while the study is ambitious in its conceptual framing (Social Cognitive Theory, Dunning–Kruger Effect, and the Two-Step Flow of Communication Model), the empirical execution remains modest. The data are based on self-reported surveys with limited knowledge items, and the operationalization of “opinion leadership” is problematic in ways that weaken the conclusions.

Response: We agree with this assessment and have revised the manuscript to ensure that the scope of empirical claims matches the data. We now consistently refer to “perceived opinion leadership” rather than actual influence, and we explicitly discuss the limitations of self-reported leadership measures (Section 2.2.2; page 15, l. 339ff -tracked version). In particular, we clarified that the study captures confidence calibration rather than actual information dissemination or behavioral influence.

Comment 2.3: In its current form, the manuscript reads as a solid empirical exercise, but one that overstates its implications and does not yet deliver the strong evidence that its theoretical framing promises. I would encourage the authors to simplify their presentation, report their results more rigorously, and be more cautious in drawing societal and policy conclusions

Response: We revised the Discussion and Conclusions to explicitly avoid causal or policy-prescriptive claims. Societal relevance is now framed in terms of communication risks and literacy challenges rather than direct policy effects (see p. 37, l. 855ff; p. 38, l. 896ff; tracked version).

Comment 2.4: “The short title is clearer and more effective than the long version.”

Response: We agree and retained the concise framing while streamlining the long title to avoid redundancy

Comment 2.5: The abstract is descriptive and clear but lacks quantitative information. Including numbers such measures of over- and underconfidence, average correct response rate of 64%, and proportion of overconfident respondents would substantiate the claims

Response: We revised the complete abstract to include key quantitative indicators, such as average knowledge accuracy and the prevalence of overconfidence (Abstract, p. 1–2 tracked version).

Comment 2.6: The abstract does not sufficiently explain why the findings matter for society. Overconfidence among opinion leaders could translate into misinformation risks and resistance to sustainable agricultural policies, but this connection is only implied, not spelled out.

Response: We agree and strengthened the abstract by explicitly linking overconfidence among opinion leaders to potential misinformation risks and challenges for evidence-based agricultural communication. Now, we explicitly link overconfidence among opinion leaders to misinformation risks and resistance to evidence-based communication in the abstract. (latter part of abstract).

Comment 2.7: The conclusion of the abstract is weak, falling back on the generic “further research is needed.” A stronger closing would highlight what is at stake if overconfidence among opinion leaders is left unaddressed.

Response: We replaced the generic closing with a more concrete statement emphasizing the risks of unaddressed overconfidence and the relevance for communication strategies. (last sentence of the abstract).

Comment 2.8: The introduction is educational and well written, but excessively long on theory. Extended explanations of SCT, DKE, and TSFCM are useful for students, but journal readers may find this overwhelming.

Response: We streamlined the Introduction by reducing explanatory passages and integrating theory more directly with the research problem and objectives (pp. 2–5, comparison with original version shows reduction)- Nevertheless, the requests of the other two reviewers to take certain theories and concepts into account were also addressed.

Comment 2.9: Objectives are only listed at the very end, which dilutes the focus. They should be stated earlier to guide the reader.

Response: We integrated the study objectives into the argumentative flow of the Introduction rather than only listing them (p.8, l. 176ff)

Comment 2.10: The scope of analysis is modest (agricultural knowledge in Germany), yet the paper mobilizes references from health, climate, nutrition, and food literacy. This gives breadth, but also raises expectations that the empirical part cannot match.

Response: We narrowed the framing language and explicitly contextualized references from adjacent domains as illustrative rather than directly tested (see p.5 l.108ff and section 3.1.3 -tracked version)

Comment 2.11: Opinion leadership is measured entirely by self-report (adapted Childers scale). This is a serious limitation: people may overestimate their influence, confusing talkativeness or ego with actual impact. Without external validation (peer nominations, network analysis, behavioral measures), the OLI remains a proxy for perceived leadership rather th

---

## [Decision Letter · Decision Letter 1]

7 Jan 2026

Do opinion leaders know more? Knowledge accuracy, self-confidence, and media use in agricultural issuess

PONE-D-25-28948R1

Dear Dr. Gabriel,

We’re pleased to inform you that your manuscript has been judged scientifically suitable for publication and will be formally accepted for publication once it meets all outstanding technical requirements.

Kind regards,

Pierluigi Vellucci

Academic Editor

PLOS One

Additional Editor Comments (optional):

Reviewers' comments:

Reviewer's Responses to Questions

**Comments to the Author**

Reviewer #2: All comments have been addressed

Reviewer #3: All comments have been addressed

2. Is the manuscript technically sound, and do the data support the conclusions?

Reviewer #2: Yes

Reviewer #3: Yes

3. Has the statistical analysis been performed appropriately and rigorously?

Reviewer #2: Yes

Reviewer #3: Yes

4. Have the authors made all data underlying the findings in their manuscript fully available?

Reviewer #2: Yes

Reviewer #3: Yes

5. Is the manuscript presented in an intelligible fashion and written in standard English?

Reviewer #2: Yes

Reviewer #3: Yes

Reviewer #2: After careful consideration of the revised manuscript and the authors’ responses to reviewers, I recommend acceptance for publication. The authors have engaged constructively and thoroughly with the peer-review process, resulting in a clearer, more rigorous, and more transparent manuscript. The revised version shows careful attention to conceptual boundaries, technical accuracy, and interpretive caution. In particular, the explicit framing of opinion leadership as perceived opinion leadership, together with the clearer presentation of the main empirical findings, substantially improves the internal coherence of the study.

While the contribution is intentionally modest and primarily descriptive, the manuscript addresses a relevant and timely intersection between communication, cognition, and agriculture, a domain that remains underrepresented in empirical research. Within this scope, the study constitutes a useful and well-executed contribution to the literature on agricultural communication and public understanding of science.

Reviewer #3: I agree with the publication of the revised version of the manuscript pending one minor adjustment: Third Person Effect and Spiral of Silence are separate models/theories and cannot be referenced together. Authors should properly reference TPE or remove mentions to TPE in the manuscript as the arguments on Motivated Reasoning seem to function properly and MR and TPE overlap.

**Do you want your identity to be public for this peer review?** For information about this choice, including consent withdrawal, please see our Privacy Policy

Reviewer #2: **Yes:** Gabriel Alves

Reviewer #3: **Yes:** João de Deus Barreto Segundo

---

## [Editor Report · Acceptance letter]

PONE-D-25-28948R1

PLOS One

Dear Dr. Gabriel,

I'm pleased to inform you that your manuscript has been deemed suitable for publication in PLOS One. Congratulations! Your manuscript is now being handed over to our production team.

Kind regards,

on behalf of

Dr. Pierluigi Vellucci

Academic Editor

PLOS One